# Efficient pure blue hyperfluorescence devices utilizing quadrupolar donor-acceptor-donor type of thermally activated delayed fluorescence sensitizers

Hyuna Lee[1,2], Ramanaskanda Braveenth [1,2], Subramanian Muruganantham [1], Chae Yeon Jeon[1], Hyun Seung Lee[1] & Jang Hyuk Kwon [1,2] ✉

The hyperfluorescence (HF) system has drawn great attention in display technology. However, the energy loss mechanism by low reverse intersystem crossing rate ($k_{RISC}$) and the Dexter energy transfer (DET) channel is still challenging. Here, we demonstrate that this can be mitigated by the quadrupolar donor-acceptor-donor (D-A-D) type of thermally activated delayed fluorescence (TADF) sensitizer materials, DBA-DmICz and DBA-DTMCz. Further, the HF device with DBA-DTMCz and $v$-DABNA exhibited 43.9% of high maximum external quantum efficiency ($EQE_{max}$) with the Commission Internationale de l'Éclairage coordinates of (0.12, 0.16). The efficiency values recorded for the device are among the highest reported for HF devices. Such high efficiency is assisted by hindered DET process through i) high $k_{RISC}$, and ii) shielded lowest unoccupied molecular orbital with the presence of two donors in D-A-D type of skeleton. Our current study provides an effective way of designing TADF sensitizer for future HF technology.

The thermally activated delayed fluorescence (TADF)-based organic light-emitting diodes (OLEDs) showed great potential in recent display technologies due to their ~100% of internal quantum efficiency (IQE). Such achievement is possible when the dark triplet excitons convert into radiative singlet excitons through an appropriate mechanism of reverse intersystem crossing (RISC) process[1–6]. Which process can be efficiently activated with the help of small singlet ($S_1$) and triplet ($T_1$) energy gap ($\Delta E_{ST}$). In general, the basic design strategy to achieve small $\Delta E_{ST}$ is depending on the spatial separation of the highest occupied molecular orbital (HOMO) and the lowest unoccupied molecular orbital (LUMO) in a donor (D) and acceptor (A) comprising vertical molecular configuration[7–9]. However, such D-A type molecules possess strong intra-charge transfer (ICT) properties with the broadened full width at half maximum (FWHM), which limits their applications in display technology[10,11]. Thus, Adachi et al. proposed the TADF sensitizer technology so-called

hyperfluorescence (HF) system to achieve high color purity and 100% of exciton utilization efficiency in OLEDs[12,13]. In a HF system, TADF materials act as exciton sensitizers and up-convert the 75% of electrically generated triplet excitons into singlet excitons. Then, the final emitter can theoretically harvest 100% of excitons through the long-range Förster resonance energy transfer (FRET) process. However, the Dexter energy transfer (DET) from the triplet state of TADF to that of final emitter also can be occurred, which is an energy loss mechanism in HF system. Thus, much effort has been made to enhance the FRET while diminishing the DET process between the TADF and final emitter in HF system. The FRET process is basically an emission light-sensitive dipole-dipole interaction, which requires large spectral overlap between the absorption spectrum of the final emitter and the emission spectrum of TADF material[14,15]. While the DET is a direct electron-exchange process and demands a short range of distance for achieving the molecular orbital overlapping[16,17].

[1]Organic Optoelectronic Device Lab (OODL), Department of Information Display, Kyung Hee University, 26, Kyungheedae-ro, Dongdaemun-gu, Seoul 02447, Republic of Korea. [2]These authors contributed equally: Hyuna Lee, Ramanaskanda Braveenth. ✉e-mail: jhkwon@khu.ac.kr

Since the DET is an energy loss process, many strategies were implemented to alleviate this DET process without harming FRET efficiency. Firstly, many researchers introduced the electrically inert bulky groups in TADF sensitizers or final emitters to prohibit the molecular orbital interactions[18–20]. In 2018, Duan et al. substituted the methyl, *tert*-butyl, and butyl-phenyl moieties at the terminal edge of the green fluorescence material, PAD. As the bulkiness of inert group was increased, the DET process was reduced, and the HF device efficiency was enhanced[21]. Further, Su et al. introduced an inert phenyl-fluorene terminal to the PXZ-DBPZ TADF sensitizer and developed FPXZ-DBPZ emitter[22]. When 0.6% of red fluorescence emitter of DBP was doped with FPXZ-DBPZ in an HF system, slower rate constant of DET ($k_{DET}$) was observed compared to that of parent PXZ-DBPZ TADF sensitizer. In addition to that reducing the orbital overlap by inert group substitution, Lee et al. proposed that the fast rate constant of RISC ($k_{RISC}$) of TADF sensitizer can lead the majority of triplet excitons to be up-converted to the singlet excitons rather than transfer to the DET channel[23]. Thus, the $k_{DET}$ to final emitter can be lowered. For example, 12BTCzTPN exhibited almost 4 times faster $k_{RISC}$ compared to 4CzTPN, and the $k_{DET}$ with 0.7% of red fluorescence emitter 4tBuMB was decreased from 1.02 to $0.53 \times 10^5 \, s^{-1}$,[24]. Moreover, Howard et al. demonstrated that the multi-donor shielding in 4CzIPN restricts the LUMO-LUMO overlap between the TADF materials, and the triplet exciton transfer can be limited[25]. Thus, TADF architecture with shielded LUMO and fast $k_{RISC}$ has the advantage of achieving an efficient HF system by reducing the DET process.

To obtain an efficient HF system, D-A-D molecular skeleton-based TADF sensitizers become promising candidates to satisfy the aforementioned requirements. The quadrupolar D-A-D type of TADF materials has been reported to have high $k_{RISC}$ with the support of large spin-orbit coupling matrix element (SOCME) values[26,27]. Among many studies, Wang et al. reported three D-A-D type of blue TADF materials, *m*-Ac-DBNA, *p*-Ac-DBNA, and *m'*-Ac-DBNA with good TADF performances[28]. Unfortunately, they exhibited sky blue emission around 500 nm due to the strong electron-donating property of acridine donor. Yasuda et al. suggested that the quadrupolar D-A-D type of TADF material can achieve additional SOCME values with the assistance of quasi-degenerate HOMO energy level and dual CT states[29]. Therefore, their reported D-A-D type of TAZ-2 showed superior performances compared to the conventional D-A type of TAZ-1. In addition, the same group reported an enhanced blue quadrupolar D-A-D type of TADF materials, QBO and QXT[30]. The QBO exhibited the maximum quantum external efficiency ($EQE_{max}$) of 24.9% with Commission Internationale de l'Éclairage (CIE) coordinate of (0.16, 0.30). From the above-reported studies, it speculates that suitable donor and acceptor moieties combination in D-A-D molecular skeleton is helpful to develop worthy TADF sensitizer candidates for pure blue emitting HF-OLEDs.

In this study, we report two quadrupolar types of blue TADF materials with D-A-D skeleton, 3,11-bis(5-phenylindolo[3,2-a]carbazol-12(5H)-yl)−5,9-dioxa-13b-boranaphtho[3,2,1-de]anthracene (DBA-DmICz) and 3,11-bis(1,3,6,8-tetramethyl-9H-carbazol-9-yl)−5,9-dioxa-13b-boranaphtho[3,2,1-de]anthracene (DBA-DTMCz). According to the detailed theoretical and photophysical investigation, we have confirmed that both emitters revealed high $k_{RISC}$ due to quasi-degenerated HOMO energy levels, and enhanced SOCME. Moreover, the TADF, as well as HF device performances using our TADF materials, are analyzed to understand their potentials in real device applications. Particularly, the HF device based on DBA-DTMCz and 1% of *v*-DABNA[31] achieved $EQE_{max}$ of 43.9% with the CIE of 0.16. Such high device efficiency is supported by the limited triplet excitons transfer to the *v*-DABNA with the assistance of D-A-D structure and fast $k_{RISC}$ of TADF sensitizers.

## Results

### Design strategy and theoretical calculation

As the desire of obtaining high FRET efficiency with minimal DET channel, TADF sensitizers are required to have high $k_{RISC}$ and shielded LUMO distribution with proper blue emission wavelength. Considering the above concept, quadrupolar D-A-D architecture in boron-based deep blue TADF materials seems to be a promising strategy. Such architecture is expected to have high $k_{RISC}$ due to the quasi-degenerated HOMO energy levels and enhanced SOCME values. In addition, the electrons in LUMO hardly move to final emitter via DET channel due to the multiple donor shielding effect. The oxygen-bridged boron type of 5,9-dioxa-13b-boranaphtho[3,2,1-de]anthracene (DBA) acceptor is selected due to its symmetrical structure with a central boron core, high PLQY, and weak accepting properties[32–35]. In order to obtain deep blue emission, weak donating property of carbazole derivatives, 5-phenyl-5,12-dihydroindolo[3,2-a]carbazole (mICz) and 1,3,6,8-tetramethyl-9H-carbazole (TMCz) are selected, and substituted at the *para* positions of boron atom of DBA acceptor in D-A-D type of molecular skeleton. Thus, the two TADF sensitizer materials of DBA-DmICz and DBA-DTMCz are designed. In order to confirm the electronic properties of the designed TADF materials, density functional theory (DFT) calculation for the optimization of the ground state, and time-dependent DFT (TD-DFT) simulations for the excited-state optimization proceeded with the Lee-Yang-Parr correlation function (B3LYP) and the basis set of 6−31(G)* using Schrödinger 2022-1 program[36,37].

As shown in Fig. 1, the LUMO is mainly localized on the DBA acceptor, while the HOMO is distributed over the peripheral donors, respectively. In addition, there is a small orbital overlap exhibiting, which is HOMO → LUMO of ICT characteristics of singlet emission ($^1CT$). The calculated LUMO energy levels of DBA-DmICz and DBA-DTMCz are 2.877 and 2.882 eV, respectively. Interestingly, both TADF materials have energetically similar HOMO and HOMO-1 energy levels, and the orbitals are localized on individual mICz and TMCz donors. The HOMO and HOMO-1 energy levels of DBA-DmICz are 5.662 and 5.682 eV, and those of DBA-DTMCz are 5.647 and 5.658 eV. Such quasi-degenerate HOMO and HOMO-1 can contribute to the ICT transition of $S_1$ and $S_2$ in the TADF process. The calculated $S_1/T_1$ energy levels were 2.888/2.749 eV and 2.808/2.798 eV, respectively. Thus, the following $\Delta E_{ST}$ values are calculated to be 0.139 and 0.010 eV, respectively. The calculated data indicates that both materials will have efficient TADF performances with blue emission.

Since the excited-state symmetry in quadrupolar D-A-D material became broken, the electronic transition state can be distributed on one of the dual charge transfer states within the material[30]. Thus, two quasi-degenerated excited states, $CT_I$ and $CT_{II}$ can be generated. Such dual-degenerate charge transfer excited states can enhance the SOCME, $<S_n|\hat{H}|T_m>$ and such enhanced SOCME can accelerate the $k_{RISC}$ of TADF materials. This can be explained by the reported Marcus theory formulated rate constant Eq. (1)[38–40]:

$$k_{RISC} = \frac{2\pi}{\hbar} |S|\hat{H}_{SOC}|T|^2 \frac{1}{\sqrt{4\pi\lambda_{RISC}k_B T}} e^{\left(\frac{-E_a^{RISC}}{k_B T}\right)} \quad (1)$$

Where, $\lambda_{RISC}$ represents the reorganization energy for RISC, $k_B$ is the Boltzmann constant, and $E_a^{RISC}$ is the activation energy for RISC. Thus, the improved SOCME between the singlet and triplet state values can contribute to accelerate the spin up-conversion RISC process. Thus, the natural transition orbital (NTO) analyses and the calculation of SOCME, $<S_n|\hat{H}|T_m>$ ($n = 1$, 2 and $m = 1$, 2, 3) of two materials are proceeded. Since both D-A-D materials exhibit two degenerated hole-occupied NTO distributions, NTOs of hole can be located on the either donor moieties and provide the dual-degenerate CT states for S ($^1CT_I$ and $^1CT_{II}$) and T ($^3CT_I$ and $^3CT_{II}$) transitions. Following the El-Sayed rule, the transitions between the $<^1CT_I|\hat{H}|^3CT_I>$ and $<^1CT_{II}|\hat{H}|^3CT_{II}>$ are

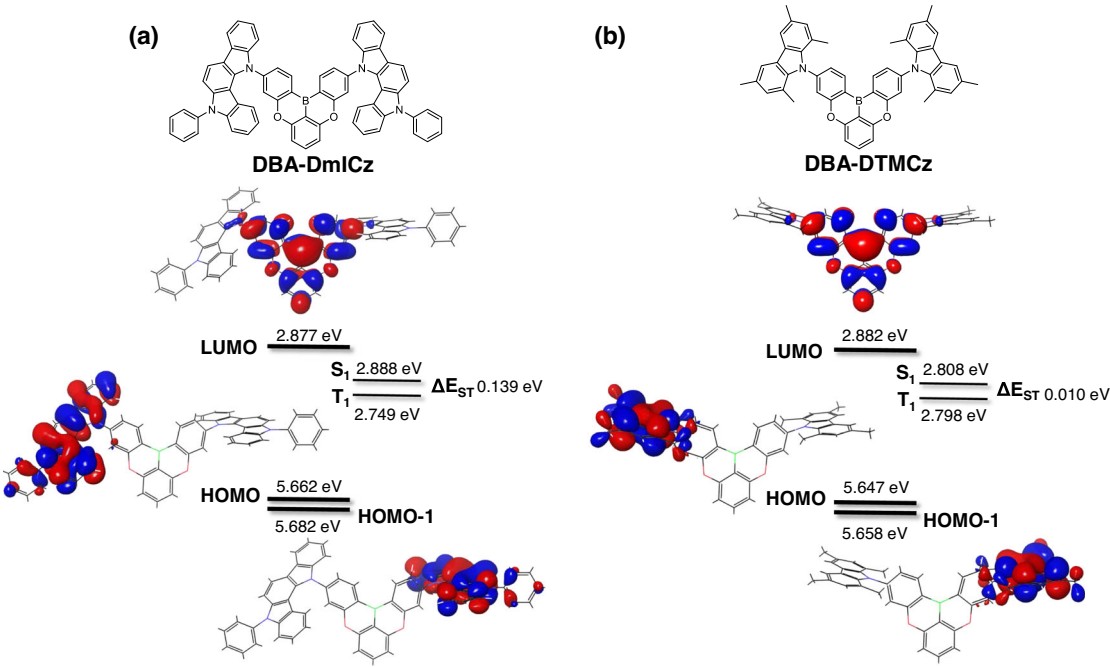

**Fig. 1 | The designed molecules and molecular simulation. Calculated energy levels. a** DBA-DmICz, and **b** DBA-DTMCz.

**Fig. 2 | Synthetic scheme.** Synthetic routes for DBA-DmICz and DBA-DTMCz.

forbidden, which of the SOCME values are small <0.02 cm$^{-1}$ as like conventional dipolar TADF materials. But, larger SOCME value can be obtained from the <$^1$CT$_I$|$\hat{H}$|$^3$LE> transition. The DBA-DmICz and DBA-DTMCz obtained enhanced value of <$^1$CT|$\hat{H}$|$^3$LE> 0.23 and 0.64 cm$^{-1}$, respectively. On the other hands, the quasi-degenerated CT states in quadrupolar system possess the large orbital angular momentum changes between <$^1$CT$_I$|$\hat{H}$|$^3$CT$_{II}$> and <$^1$CT$_{II}$|$\hat{H}$|$^3$CT$_I$>, which is allowed transition, and additional SOCME can be obtained. Thus, DBA-DmICz and DBA-DTMCz obtained the additional SOCME value involving <$^1$CT$_I$|$\hat{H}$|$^3$CT$_{II}$>, <$^1$CT$_{II}$|$\hat{H}$|$^3$CT$_I$>, and <$^1$CT$_{II}$|$\hat{H}$|$^3$LE>, and this can improve the RISC process with fast spin-flip conversion. The detailed illustrates of NTO distribution and corresponding individual SOCME values were described in Supplementary Figs. 1–2.

## Material synthesis and photophysical properties

The designed target molecules of DBA-DmICz and DBA-DTMCz are synthesized using a palladium-catalyzed Buchwald-Hartwig cross-coupling reaction under a nitrogen atmosphere. The synthetic routes of DBA-DmICz and DBA-DTMCz are depicted in Fig. 2. The detailed synthetic procedure, high-resolution mass spectra (HRMS), and nuclear magnetic resonance (NMR) spectra of compounds are described in Supplementary Figs. 18–33.

The fundamental photophysical properties of DBA-DmICz and DBA-DTMCz are measured in toluene solution and 30% doped dibenzo[*b,d*]furan-2,8-diylbis(diphenylphosphine oxide) (DBFPO) film. The ultraviolet-visible (UV-vis) absorption and photoluminescence (PL) spectra are recorded at 300 K in toluene solution. The

phosphorescence spectra are measured in toluene solution at 77 K with 30 ms of delaying time from the initial excitation. In UV-vis absorption, the absorption peaks of carbazole derivatives are recorded ~350 nm, and ICT induced n-π* absorption peaks of DBA-DmICz and DBA-DTMCz appeared at 394 and 390 nm, respectively. The maxima of PL emissions in the steady state are noticed at 448 and 455 nm, respectively. As expected from the simulation results, both materials showed deep blue emissions. They exhibited the featureless and broad shape of PL spectra, which is evidence of typical CT emission. Interestingly, the shapes of phosphorescence spectra are different, which indicates that the triplet orbital densities are not same between these two materials. Thus, the phosphorescence spectra of individual moieties are measured, and placed in Supplementary Fig. 3. It is noticeable that the low concentration state of phosphorescence spectrum shows the $^3$LE state, which of DBA-DmICz was mainly derived from the mICz donor moiety with DBA acceptor, while that of DBA-DTMCz was combined of the DBA acceptor and TMCz donor moiety. The T$_1$ energy level is determined from the onset of phosphorescence spectra, and the T$_1$ values of DBA-DmICz and DBA-DTMCz are 2.96 and 3.00 eV, respectively. Moreover, the ΔE$_{ST}$ values are calculated to be 0.12 and 0.02 eV, respectively, which are small enough to have an efficient RISC process. All the detailed photophysical and electrochemical data are summarized in Table 1. All the measurement graphs of electrochemical and thermal properties are illustrated in Supplementary Figs. 4, 5. In order to evaluate the TADF performances of DBA-DmICz and DBA-DTMCz, the measurements of absolute PLQY values and transient PL (TRPL) characteristics are proceeded using 30% doped in DBFPO host

**Table 1 | The photophysical and electrochemical properties of DBA-DmICz and DBA-DTMCz**

| | $\lambda_{abs}$ (nm)[a] | $\lambda_{RTPL}$ (nm)[b] | FWHM (nm)[b] | $S_1$ (eV)[c] | $T_1$ (eV)[d] | $\Delta E_{ST}$ (eV)[e] | $\tau_p$ (ns)[f] | $\tau_d$ (μs)[f] | HOMO (eV)[g] | LUMO (eV)[h] | PLQY[i] |
|---|---|---|---|---|---|---|---|---|---|---|---|
| DBA-DmICz | 394 | 448 | 58 | 3.08 | 2.96 | 0.12 | 37.3 | 1.94 | 5.85 | 2.87 | 0.95 |
| DBA-DTMCz | 390 | 455 | 53 | 3.02 | 3.00 | 0.02 | 36.0 | 0.92 | 5.82 | 2.89 | 0.99 |

[a]UV-Vis absorption wavelengths measured in $10^{-5}$ M of toluene solution. [b]The maximum photoluminescence emission measured in toluene at 300 K. [c]Calculated by the onset of PL spectrum in toluene at 300 K. [d]Calculated by the onset of phosphorescence spectrum in toluene at 77 K with 30 ms of delaying. [e]$\Delta E_{ST} = S_1 – T_1$. [f]Prompt and delayed lifetime in 30% doped DBFPO film. [g]HOMO level energy measured by cyclic voltammetry (CV) method. [h]LUMO = HOMO-optical band gap. [i]The absolute PLQY value measured in 30% doped DBFPO film.

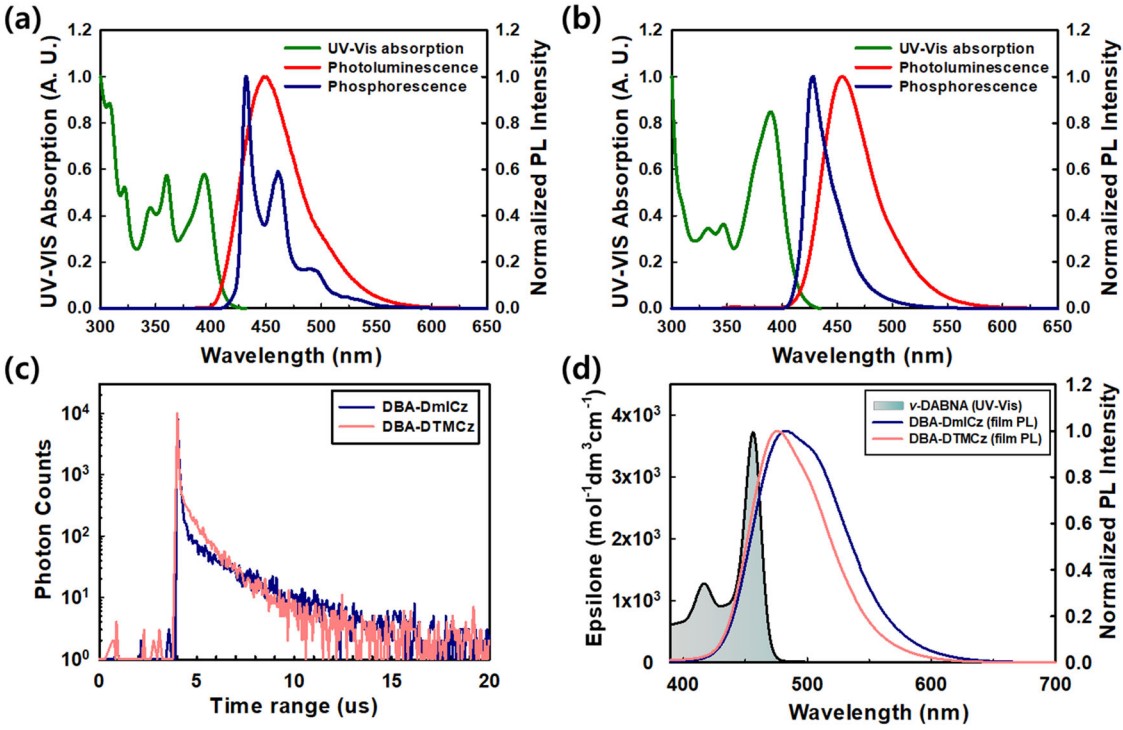

**Fig. 3 | The photophysical measurements.** The normalized UV-Vis absorption, PL, and phosphorescence spectra of **a** DBA-DmICz and **b** DBA-DTMCz in toluene solution ($10^{-5}$ M). The UV-Vis absorption and PL spectra were measured in 300 K, and the phosphorescence spectra were measured in 77 K with 30 ms of delaying. **c** TRPL measurement in 30% doped film of DBFPO host matrix under taken photoexcitation at 365 nm. **d** Spectrum overlapping between normalized 30% doped film PL emissions and absorption epsilon values of v-DABNA.

film as shown in Fig. 3c. In film PL state, both exhibited broaden and bathochromic shifted emissions which can be derived of high concentration induced-aggregation effect, and stabilization effect in polar DBFPO host medium. Despite of high doping concentration of 30%, the absolute PLQY values of DBA-DmICz and DBA-DTMCz are maintained high as 0.95 and 0.99, respectively.

As shown in Fig. 3c, both doped films exhibited clear exponential decay curves comprising of the nano-second scale of prompt, and the micro-second scale of delayed fluorescence emission. Further, to figure out TADF characteristics of the delayed decay curves, temperature-dependent TRPL measurements proceeded as shown in Supplementary Fig. 6. As the temperature became lower, the delayed portion is decreased, which confirms that delayed decays are supporting the TADF characteristics. The delayed exciton lifetimes ($\tau_d$) of DBA-DmICz and DBA-DTMCz were 1.94 and 0.92 μs, which are in very short range among reported deep blue TADF materials. The rate constants of radiative decay ($k_r$), intersystem crossing ($k_{ISC}$), and $k_{RISC}$ are calculated to quantify the TADF performances of both emitters using the previously reported method[41]. The individual excited-state rate constant values are summarized in Supplementary Table 1. With high PLQY values, DBA-DmICz and DBA-DTMCz exhibited the 1.87 and $1.42 \times 10^7$ s$^{-1}$ of high $k_r$ values, respectively. Although the $k_{ISC}$ values are also high as 0.81 and $1.37 \times 10^7$ s$^{-1}$, and their $k_{RISC}$ values are calculated

as 0.62 and $2.10 \times 10^6$ s$^{-1}$, respectively. Such high $k_{RISC}$ values are attributed to small $\Delta E_{ST}$ and enhanced SOCME from degenerated dual CT transitions as expected in the NTO calculation. To verify such high $k_{RISC}$ values, we confirmed the characteristics of triplet state in 30% doped film state. Both materials exhibited a clear $^3$CT characteristics of phosphorescence spectra as shown in Supplementary Fig 7. The $\Delta E_{ST}$ values in film states of DBA-DmICz and DBA-DTMCz are 0.05 and 0.01 eV, respectively. For efficient RISC process, strong vibronic coupling (VC) between the $^3$LE and $^3$CT should be involved. With closely lying excited states, $\Delta E_{3LE-3CT}$ values are 0.07 and 0.06 eV, respectively, which is small enough to efficiently induce the VC. Additionally, large SOCME values between the dual $^1$CT and $^3$LE accelerate the efficient RISC procedure. To prove the enhanced RISC process of both materials, $k_{RISC}$ values of D-A skeleton of TADF materials with the same acceptor and donor moiety are investigated. The theoretical and photophysical properties of 9-(5,9-dioxa-13b-boranaphtho[3,2,1-de]anthracen-7-yl)–1,3,6,8-tetramethyl-9H-carbazole (DBA-TMCz) was reported by Adachi et al. using the name of TMCz-BO[42]. And the 12-(5,9-dioxa-13b-boranaphtho[3,2,1-de]anthracen-7-yl)−5-phenyl-5,12-dihy-droindolo[3,2-a]carbazole (DBA-mICz) was synthesized as shown in Supplementary Fig. 33. The detailed structural analysis of DBA-mICz is depicted in Supplementary Figs. 25–27. All the measured photophysical properties and the calculated rate constant values are

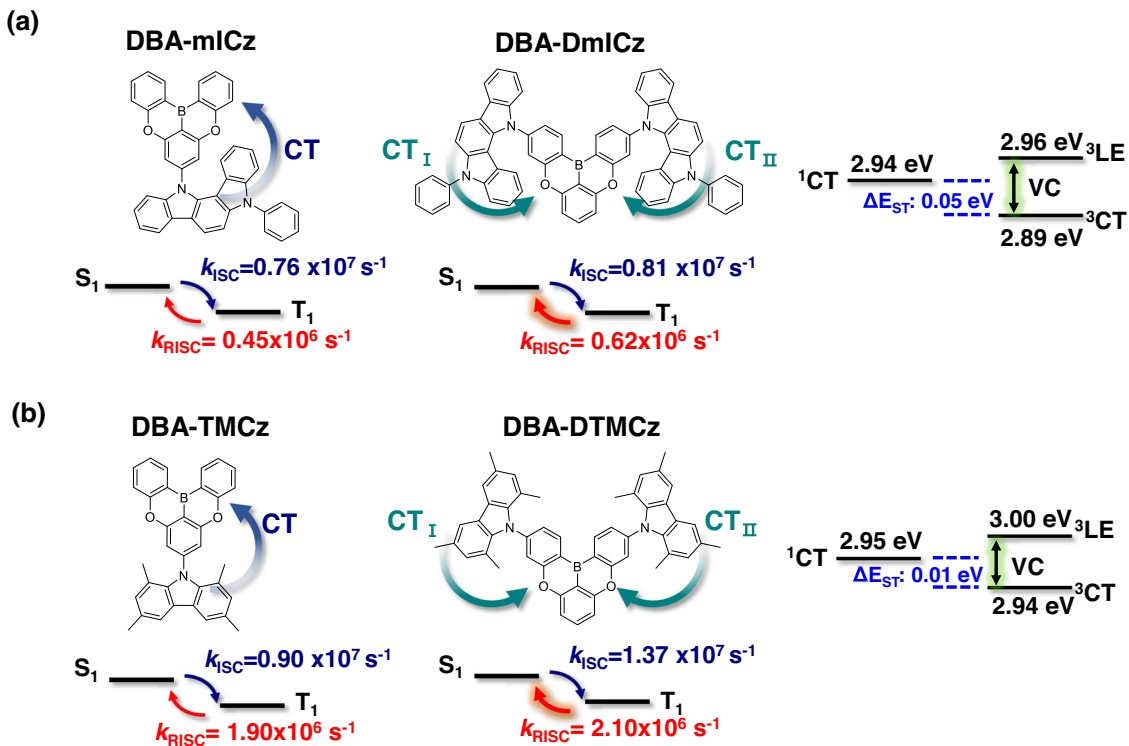

**Fig. 4 | Schematic illustration of TADF mechanism.** The energy levels and comparison of RISC and ISC rate constants between bipolar D-A and quadrupolar D-A-D types **a** DBA-mICz and DBA-DmICz, **b** DBA-TMCz and DBA-DTMCz. The energy levels and rate constants were measured in 30% doped films. The rate constants of DBA-TMCz were reported by Adachi et al. using the name TMCz-BO[42].

illustrated in Supplementary Fig. 8 and Supplementary Tables 2, 3, respectively.

As expected, both TMCz-BO and DBA-mICz materials exhibited lower $k_{RISC}$ values compared to those of DBA-DmICz and DBA-DTMCz as shown in Fig. 4. The reported $k_{RISC}$ value of 30% of TMCz-BO in DBFPO host was $1.90 \times 10^6$,[42] and that of DBA-mICz is $4.50 \times 10^5 \, s^{-1}$, respectively. The higher intensity of ICT absorption peak of DBA-DmICz in Supplementary Fig. 8d indicates that the dual ICT transitions also can accelerate the RISC procedure through strengthened CT transitions and SOCME. Subsequently, the intensified ICT absorption can enhance the prompt and total PLQY of DBA-DmICz compared to DBA-mICz[29]. The prompt PLQY increased from 64 to 70%, and the total PLQY enhanced from 85 to 95%, respectively. Considering that the mostly reported $k_{DET}$ range of ~$10^5 \, s^{-1}$, the higher $k_{RISC}$ value of both materials can be anticipated to lead triplet excitons to be up-converted rather than transfer to DET channel.

**Device performance**

In order to evaluate the TADF performances of synthesized materials, TADF-OLED devices are fabricated with previously reported optimized device architecture[32–35]; ITO (50 nm)/HATCN (7 nm)/TAPC (55 nm)/DCDPA (10 nm)/DBFPO: 30% of DBA-DmICz and DBA-DTMCz (25 nm)/DBFPO (10 nm)/TPBi (20 nm)/LiF (1.5 nm)/Al (100 nm) were fabricated. Indium-tin-oxide (ITO) and aluminum (Al) are utilized as anode and cathode, respectively. Dipyrazino[2,3-f:2′,3′-h]quinoxaline-2,3,6,7,10,11-hexacarbonitrile (HATCN) and lithium fluoride (LiF) acted as the hole- and electron injection layers, respectively. 1,1-Bis[4-[N,N′-di(p-tolyl)amino]-phenyl] cyclohexane (TAPC) and 1,3,5-tris(1-phenyl-1H-benzo[d]imidazole-2-yl)benzene (TPBi) served as hole- and electron-transporting layers, respectively. 3,5-Di(9H-carbazol-9-yl)-N,N-diphenylaniline (DCDPA), and DBFPO were utilized as electron and hole blocking layers, respectively, on each sides of the EML with their high triplet energy levels. The DBFPO also served as host material.

All the plots of the current density-voltage-luminance (J-V-L) characteristics, external quantum efficiency versus luminance (EQE-L), and the electroluminescence (EL) spectra at 1000 cd/m² are illustrated in Fig. 5. Other device performances with different doping concentrations are shown in Supplementary Fig. 9 and Supplementary Table 4. The DBA-DmICz and DBA-DTMCz exhibited maximum EL emissions of 485 and 479 nm at 1000 cd/m², and the corresponding CIE chromaticity coordinates are (0.16, 0.34) and (0.14, 0.27), respectively. Although the solution PL of DBA-DmICz in toluene emitted 7 nm of deep blue color, the EL spectrum in polar host medium of DBFPO revealed further bathochromic shifted emission due to stronger stabilization characteristics. As shown in Supplementary Fig. 10, DBA-DmICz exhibited a strong solvatochromic effect and ICT characteristics. The maximum PL emission shifting of DBA-DmICz from n-hexane to methylene chloride is 120 nm, while DBA-DTMCz is 102 nm in the same medium. The fabricated TADF device performances are summarized in Table 2. The DBA-DmICz-based TADF device exhibited $EQE_{max}$ of 31.6%, and remained 29.3% at 1000 cd/m². Subsequently, DBA-DTMCz-based TADF device revealed a significantly high $EQE_{max}$ of 37.0, and 35.5% at 1000 cd/m², which only 4% of EQE was reduced from the $EQE_{max}$. This can be attributed to short triplet exciton decay lifetime, which effectively reduces the triplet exciton accumulation, triplet-triplet annihilation (TTA), and triplet-polaron annihilation (TPA) process[43–46]. In addition, such high device efficiency value over 30% could be obtained with the help of high PLQY, excellent TADF performance, and intensified out-coupling efficiency ($n_{out}$) from high horizontal orientation factor ($\Theta$). Since the light mainly emits and propagates perpendicular to the direction of transition dipole moment of emitter, light out-coupling efficiency can be affected depending on the $\Theta$ value[47,48]. Thus, the $\Theta$ measurement is proceeded using the previously reported method[49]. In general, high light out-coupling can be obtained when the direction of transition dipole moment (TDM) is aligned horizontally with the plane of the material[50]. And the direction

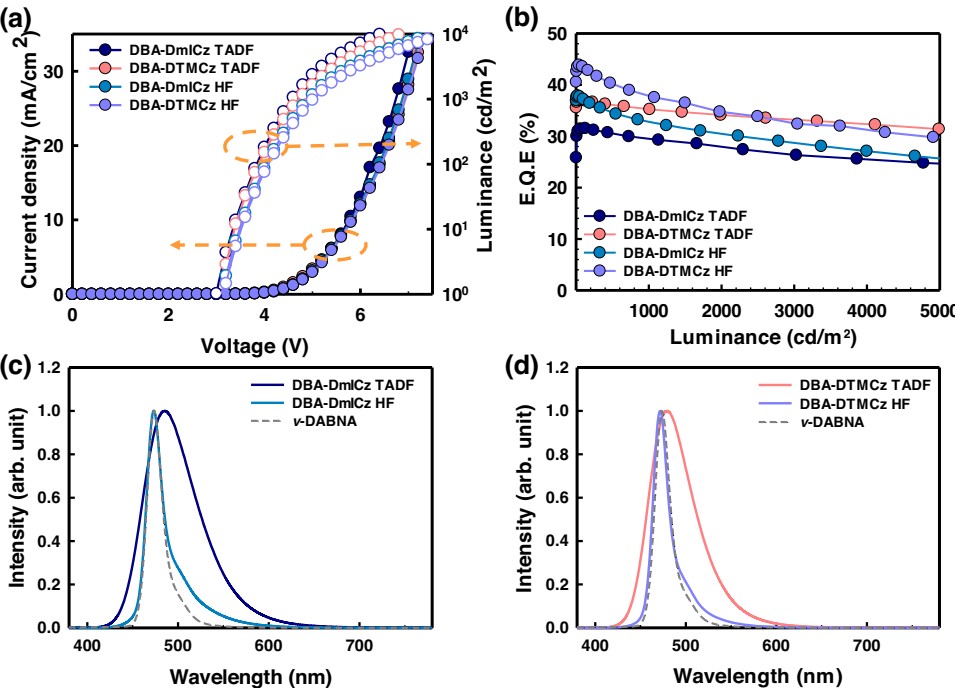

**Fig. 5 | The device performances. a** J-V-L curves, and **b** EQE versus luminance. The EL spectra of TADF and HF devices of **c** DBA-DmICz and **d** DBA-DTMCz with 1% of v-DABNA.

of TDM often closely coincides with the longest axis of donor and acceptor skeleton in TADF material[51]. Thus, the TDMs of DBA-DmICz and DBA-DTMCz are studied to confirm the high $\Theta$ value. As expected, the direction of TDM of DBA-DTMCz lied parallel to the longest axis of D-A-D linking as illustrated in Supplementary Fig. 11b. Rather, TDM of DBA-DmICz slightly passed slantwise following the slightly bent skeleton of acceptor. Thus, the lower $\Theta$ value may be obtained. As expected, DBA-DmICz and DBA-DTMCz showed high $\Theta$ of 0.79 and 0.86, respectively, as shown in Supplementary Fig. 12a, b. With 0.99 of absolute PLQY value, DBA-DTMCz able to achieve such high EQE_max value among the reported D-A-D types of blue TADF materials. In order to verify the high performance of D-A-D type of both TADF materials, the device performances of D-A type of DBA-mICz are also evaluated. In the same device configuration, 30% of DBA-mICz exhibited 18.6% of EQE_max, and the maximum EL peak was 467 nm. Thus, it can be confirmed that TADF device performance of D-A-D skeleton is better than that of D-A skeleton. The device performances of DBA-mICz are illustrated in Supplementary Fig. 13 and Supplementary Table 5. Since both materials exhibited high $\Theta$ value, the emission light in front direction became enhanced and the non-Lambertian shape of angular distribution formed. Thus, the overestimated EQE values are demanded to be re-calculated by integrating all angular-dependent emission patterns as described in Supplementary Fig. 14a[52]. The following revised EQE_max values for DBA-DmICz and DBA-DTMCz are 30.6 and 35.6%, respectively. Since both TADF materials revealed high device performances with blue color emission, the fabrication of HF devices with 1% of v-DABNA followed the same device configuration. As shown in Fig. 5c, d, HF devices with DBA-DmICz and DBA-DTMCz exhibited the same maximum EL peak of 473 nm with 21 nm of narrow FWHM. However, the achieved CIE coordinates are different, (0.13, 0.22), and (0.12, 0.16), respectively, derived of the different EL performances of two TADF materials. The DBA-DTMCz showed hypsochromic-shifted emission than DBA-DmICz, so slightly larger area of spectrum overlap is obtained as shown in Fig. 3d. Thus, higher FRET efficiency ($\Phi_{FRET}$) can be expected. In addition, the EL spectrum of HF device is composed of the emission portion of TADF and v-DABNA, narrow FWHM

can be advantageous to maintain the purity of HF spectrum. With good EL performance, the HF device efficiencies using DBA-DmICz and DBA-DTMCz are enhanced to 37.7 and 43.9%, respectively. Considering the high $n_{out}$ of v-DABNA, the EQE_max values are re-calculated to be 36.5 and 42.2%, respectively, and angular distributions of HF devices are illustrated in Supplementary Fig. 14b. Especially, DBA-DTMCz utilized HF device exhibited the maximum HF device efficiency among reported v-DABNA based HF devices up to now. Such high device efficiency improvement is supported by high PLQY, improved $\Theta$, and additional TADF performance of v-DABNA with good $\Phi_{FRET}$ as previously reported[33]. Particularly, the $\Theta$ value of v-DABNA in DBA-DTMCz based HF system is 0.95 with the intensified intermolecular interactions with DBA-DTMCz and DBFPO of binary medium. The strong intermolecular interaction between the host medium and emitter is one of the key factors to determine such high $\Theta$ value. The contributions from CH/π intermolecular interactions between the emitter and the methyl groups on carbazole in DBA-DTMCz and CH/O bond from the P = O bond in DBFPO can enhance the $\Theta$ value of v-DABNA in DBA-DTMCz and DBFPO of binary medium[53,54]. In addition, highly horizontally oriented alignment of DBA-DTMCz doped binary host medium can affect the $\Theta$ value of v-DABNA[55,56]. The theoretically possible device EQE values versus PLQY and $\Theta$ are plotted in Supplementary Fig. 15. We could see a good correlation result between the optical simulation and real device efficiency values. Unfortunately, lifetimes of the fabricated devices are very short due to the high triplet energy levels of both TADF emitters.

**Energy transfer analysis**

In order to figure out the reason of such high efficiency enhancement from TADF to HF devices, energy transfer studies are performed. Since the emission spectra of two TADF materials and v-DABNA are located on a similar wavelength region, detecting pure TADF emission is difficult. Thus, we selected the photoexcitation wavelength of 365 nm to ensure the dominant exciton behaviors from TADF materials. As mentioned above, successful HF system can be obtained with maximizing the $k_{FRET}$ while minimizing the $k_{DET}$. Thus, the $k_{FRET}$ and $k_{DET}$

**Table 2 | Device performances of 30% of DBA-DmICz and DBA-DTMCz TADF devices and HF devices with 1% of v-DABNA**

| | Turn on voltage (V)[a] | Driving voltage (V)[b] | Current Efficiency (Cd/A) (Max/1000 cd/m²) | EQE (%) (Max/1000 cd/m²) | Max emission peak (nm)[c] | FWHM (nm)[c] | CIE coordinates[c] |
|---|---|---|---|---|---|---|---|
| 30% DBA-DmICz | 3.2 | 4.9 | 58.3/53.5 (56.7/51.9) | 31.6/29.3 (30.6/28.4) | 485 | 68 | (0.16, 0.34) |
| 30% DBA-DmICz 1% v-DABNA | 3.2 | 4.9 | 44.5/40.4 (42.5/38.3) | 37.7/32.6 (36.5/31.7) | 473 | 21 | (0.13, 0.22) |
| 30% DBA-DTMCz | 3.2 | 4.7 | 50.9/46.4 (48.8/44.6) | 37.0/35.5 (35.6/34.3) | 479 | 58 | (0.14, 0.27) |
| 30% DBA-DTMCz 1% v-DABNA | 3.2 | 5.1 | 42.0/36.6 (40.1/34.8) | 43.9/37.5 (42.2/35.9) | 473 | 21 | (0.12, 0.16) |

[a]Turn on voltage at 1 cd/m², [b]Driving voltage at 1000 cd/m², [c]Measured at 1000 cd/m².

were quantitatively calculated following the Eqs. (2) and (3), respectively, with the assumption of (i) $k_{PF} \gg k_{DF}$ and (ii) $k_{r,S}$, $k_{ISC}$, $k_{FRET} \gg k_{nr,S}, k_{nr,T}, k_{r,T} k_{DET}$[57–60]

$$k_{FRET} \approx k_{PF} - k_{r,S} - k_{ISC} \tag{2}$$

$$k_{DET} \approx \frac{k_{PF}k_{DF} - k_{RISC}(k_{r,s} + k_{FRET})}{k_{r,s} + k_{ISC} + k_{FRET}} - k_{nr,T} \approx k_{DF} - k_{nr,T} - k_{RISC} + \frac{k_{RISC}k_{ISC}}{k_{PF}} \tag{3}$$

Where $k_{PF}$ and $k_{r,S}$ are prompt emission and singlet radiative rate constant, respectively. The $k_{DF}$ and $k_{nr,T}$ represents the delayed emission and non-radiative triplet decay rate constant, respectively. Although v-DABNA exhibits the TADF performance, constitutes only 1% in the entire HF system and its TADF portion is quite small. Therefore, most of the exciton behaviors reflect the TADF materials, so it can be assumed that TADF performance of v-DABNA hardly affects the calculation of $k_{DET}$. The calculated $k_{FRET}$ values of 30% of DBA-DmICz and DBA-DTMCz with 1% of v-DABNA are 3.81 and $4.26 \times 10^7 s^{-1}$, respectively. These $k_{FRET}$ values are high enough to overwhelm the $k_{r,S}$ and $k_{ISC}$ values as provided in Supplementary Table 1, and this can be a clear evidence of active FRET process in these two HF systems. In addition, the $k_{DET}$ values in the same HF systems are calculated as 1.94 and $1.70 \times 10^5 s^{-1}$, respectively, which are much lower than other rate constants values. Compared with $k_{RISC}$ values of both TADF materials, the ratio of $k_{RISC}/k_{DET}$ in DBA-DmICz utilizing HF system is 3.2, and that of in DBA-DTMCz is 12.3, respectively. Such high $k_{RISC}/k_{DET}$ of DBA-DTMCz attracts large portion of triplet excitons to be up-converted to singlet excitons rather than to be transferred to DET channel as shown in Fig. 6a. Thus, most of triplet state-populated excitons were able to be converted to singlet excitons and transferred to singlet state of final emitter via FRET channel, maximizing the IQE in the final emitter in HF device. All prompt and delayed decay curves are measured and provided in Fig. 6c, d. Interestingly, our TADF materials exhibited relatively lower $k_{DET}$ values compared to other previously reported HF system. Among them, pMDBA-DI exhibited a similar EL emission wavelength of 483 nm and similar $k_{FRET}$ of $4.14 \times 10^7 s^{-1}$. On the other hand, the $k_{DET}$ value of pMDBA-DI is higher than our materials, which are $2.23 \times 10^5 s^{-1}$, despite of the high $k_{RISC}$ of $1.14 \times 10^6 s^{-1}$[33]. In order to confirm the lower $k_{DET}$ value with DBA-DTMCz, we additionally performed the energy transfer study with 2,5,8,11-tetra-tert-butylperylene (TBPe), and compared it with D-A type of pMDBA-DI. Since the TBPe is pure fluorescent blue material, we can avoid the influence of TADF performance from final emitter. With 20% of TADF sensitizers and 1% of TBPe, the $k_{DET}$ values are obtained to 2.48 and $3.12 \times 10^5 s^{-1}$ in DBA-DTMCz and pMDBA-DI HF system, respectively. Although the absolute values are slightly increased, DBA-DTMCz exhibited a lower $k_{DET}$ value with TBPe. All the calculated TRPL results are illustrated in Supplementary Fig. 16 and Supplementary Tables 6, 7.

Such low DET process may be derived from the long intermolecular distance due to the presence of bulky moiety at the terminal site or LUMO-shielded D-A-D skeleton of DBA-DmICz and DBA-DTMCz. However, the simulation results indicate that the triplet density of both materials are distributed throughout the skeleton as shown in Supplementary Fig. 17. According to Howard et al., lack of LUMO-LUMO overlap can lower the electron charge transfer than hole 3 to 4 orders, and prohibit the transport of triplet excitons[25]. Since the LUMO distribution of both two materials was shielded by the HOMO of donors, only limited triplet excitons were able to transport via DET channel, resulting low $k_{DET}$ value as illustrated in Fig. 6b.

In order to demonstrate the limited intermolecular electron charge transfer, rate constant of concentration quenching ($k_{CQ}$) of DBA-mICz, DBA-DmICz and DBA-DTMCz are measured at 10, 30, 60,

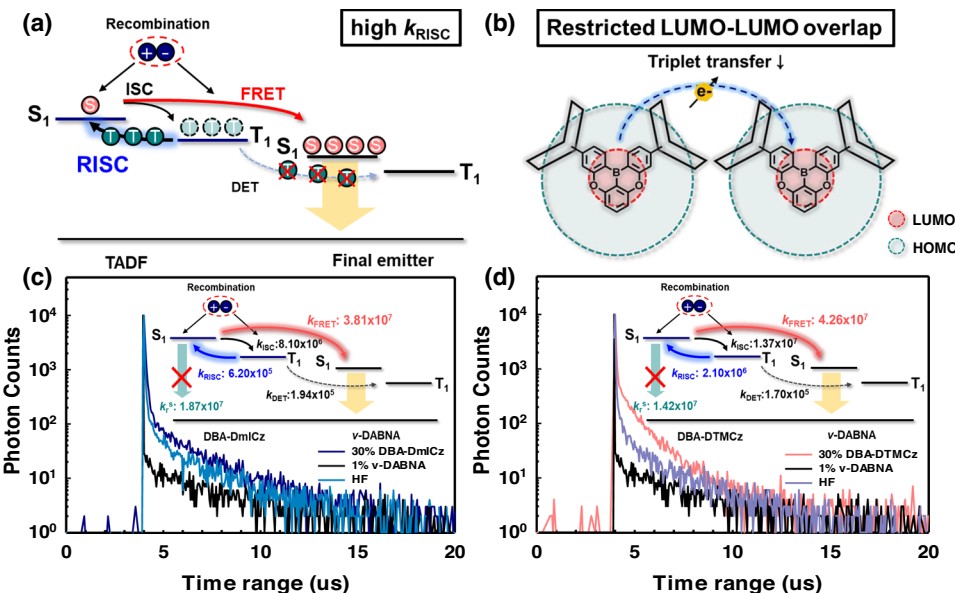

**Fig. 6 | Schematic diagram for exciton transfer in HF system.** Illustrations with respect to **a** rate constant of RISC process and **b** LUMO and LUMO overlapping. The TRPL measurement TADF, finial emitter, and HF system utilizing **c** DBA-DmICz, and

**d** DBA-DTMCz. The photoexcitation wavelength was 365 nm and the emission wavelength was detected at around 480 nm.

and 100% doped in DBFPO host film as following the reported Eq. (4):[61]

$$k_{CQ} = \frac{1}{2}\left(k_{PF} + k_{DF} - 2k_{RISC} - \sqrt{(k_{PF} - k_{DF})^2 - 4k_{ISC}k_{RISC}}\right) \quad (4)$$

Here, the $k_{CQ}$ stands for the effective intermolecular electron-exchange interaction, indicating the long triplet excitons transfer distance and active DET process. Among three TADF materials, DBA-mICz exhibited the highest $k_{CQ}$ value, which increased from 1.19 to $8.47 \times 10^5 \, s^{-1}$ from 10 to 100% doped condition. On the other hand, the DBA-DmICz showed low $k_{CQ}$ value of 0.67 to $4.15 \times 10^5 \, s^{-1}$. Especially, DBA-DTMCz showed similar $k_{CQ}$ value ~$4.25 \times 10^5 \, s^{-1}$, which is lower than that of D-A type of MCz-XT ($6.18 \times 10^5 \, s^{-1}$)[61]. Presence of same donor with similar TADF performance, D-A-D type of TADF materials exhibit low $k_{CQ}$ value, which stands for the limited DET process. And this can be due to the lack of LUMO-LUMO overlapping for electron charge transfer. All measured decay curves, PLQY are described in Fig. 7, and calculated rate constants are listed in Supplementary Table 8. Consequently, it can be demonstrated that the selecting quadrupolar D-A-D type of TADF assistant materials would be the optimum strategy for maximizing HF device efficiency via (i) achieving high $k_{RISC}$ value with the assistance of large SOCME, and (ii) lowering the electron charge transport by donor shielding LUMO structure.

## Discussion

In summary, highly efficient quadrupolar D-A-D type of blue emitting TADF materials, DBA-DmICz and DBA-DTMCz are developed and utilized as superior TADF sensitizer materials in HF system. With the assistance of two quasi-degenerated CT states, both materials are able to harvest additional SOCME value and accelerate the triplet spin-up-conversion process. Benefitting from high PLQY and high $k_{RISC}$ value, DBA-DmICz and DBA-DTMCz exhibited 31.6 and 37.0% of EQE$_{max}$ with small efficiency roll-off, respectively. Moreover, DBA-DTMCz utilized HF device with 1% of v-DABNA achieved EQE$_{max}$ of 43.9%, which is the highest value among any other reported HF devices so far. Such high-efficiency improvement is mainly due to the alleviated DET channel from high $k_{RISC}$ value and LUMO shielded structure, which is derived from quadrupolar D-A-D type of TADF material. We believe that our current findings will be helpful to design suitable TADF assistant host

materials to achieve a successful pure blue HF system for real device applications.

## Methods

### Synthesis of DBA-mICz

7-bromo-5,9-dioxa-13b-boranaphtho[3,2,1-de]anthracene DBA-Br (2 g, 5.73 mmol), 5-phenyl-5,12-dihydroindolo[3,2-a]carbazole mICz (2.09 g, 6.30 mmol), tris(dibenzylideneacetone)dipalladium(0) (0.052 g, 0.06 mmol), 2-dicyclohexylphosphino-2′,4′,6′-triisopropylbiphenyl (0.054 g, 0.12 mmol), sodium *tert*-butoxide (1.54 g, 16.04 mmol) and anhydrous toluene (20 mL) were added in to a dried two neck round bottom flask equipped with a condenser at room temperature. Then, the above mixture kept stirring at 100°C for 10 hours under an inert condition. Crude mixture was worked up using dichloromethane and water. Collected organic layer was dried over anhydrous magnesium sulfate and concentrated after the filtration. Finally, the crude mixture was precipitated using toluene and *n*-hexane. Yield: 60 %; [1]H NMR (400 MHz, CDCl₃) δ: 8.78 (d, *J* = 6.4 Hz, 2H), 8.15−8.19 (m, 2H), 7.71−7.75 (m, 2H), 7.59−7.66 (m, 5H), 7.45 (t, *J* = 7.2 Hz, 2H), 7.34−7.48 (m, 3H), 7.24−7.28 (m, 1H), 7.11 (t, *J* = 7.6 Hz, 1H), 6.59 (t, *J* = 7.6 Hz, 1H), 6.20 (d, *J* = 8.0 Hz, 1H); [13]C NMR (100 MHz, CDCl₃) δ 160.64, 158.27, 146.07, 141.75, 141.65, 140.50, 137.72, 136.48, 134.65, 133.92, 129.95, 128.07, 124.96, 124.49, 124.43, 123.43, 123.22, 121.31, 120.98, 119.65, 119.04, 118.67, 118.36, 117.98, 110.39, 109.23, 108.99, 108.18, 104.27. HRMS (ESI) *m/z*: Anal. calcd. For C₄₂H₂₅BN₂O₂, 600.2009; found, 600.2011.

### Synthesis of DBA-DmICz

3,11-dibromo-5,9-dioxa-13b-boranaphtho[3,2,1-de]anthracene DBA-p2Br (1.0 g, 2.33 mmol), 5-phenyl-5,12-dihydroindolo[3,2-a]carbazole mICz (1.63 g, 4.90 mmol), tris(dibenzylideneacetone)dipalladium(0) (0.085 g, 0.09 mmol), 2-dicyclohexylphosphino-2′,6′-dimethoxybiphenyl (0.076 g, 0.18 mmol), sodium *tert*-butoxide (0.62 g, 6.54 mmol) and anhydrous toluene (22 mL) were added in to a dried two neck round bottom flask equipped with a condenser at room temperature. Then, the above mixture kept stirring at 100°C for 8 hours under an inert condition. After complete disappearing of stating materials, crude mixture was worked

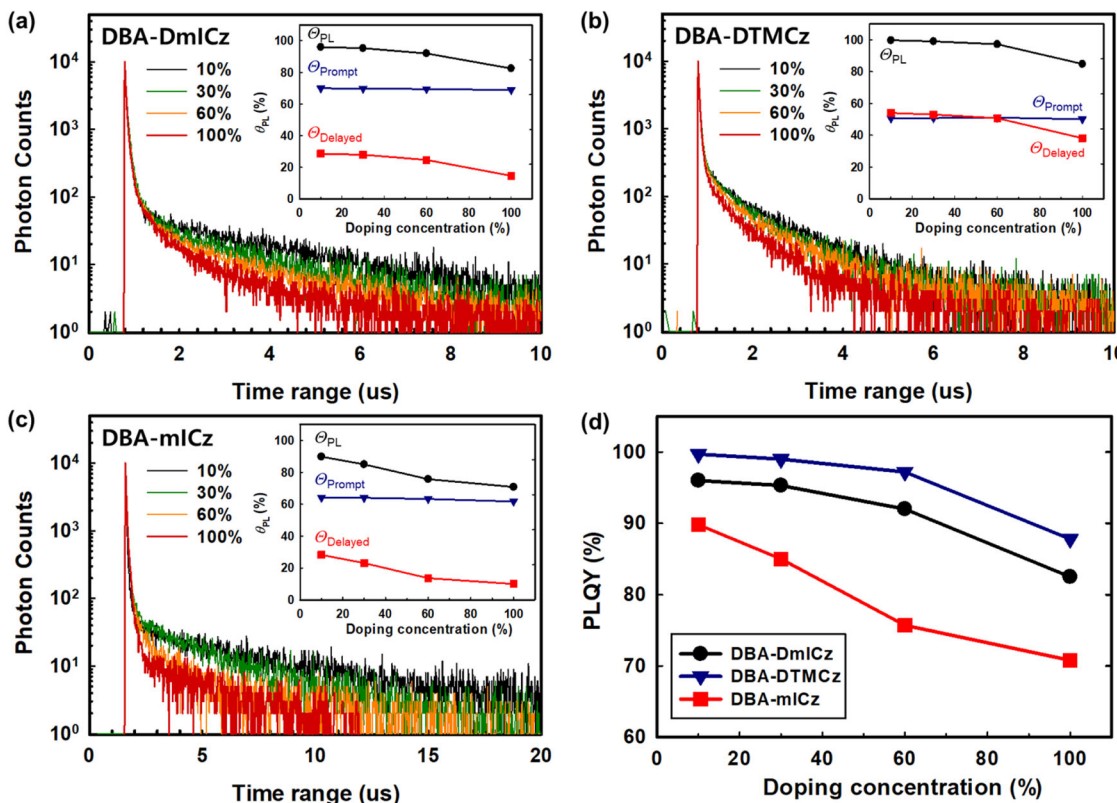

**Fig. 7 | Transient PL and PLQY.** TRPL decay curves and PLQY variation **a** DBA-DmICz, **b** DBA-DTMCz, and **c** DBA-mICz depending on the doping concentration. **d** The plot of PLQY depends on the doping concentration.

up using chloroform (30 mL*3) and water (90 mL). Collected organic layer was dried over anhydrous magnesium sulfate and filtered using a filter paper. Then, collected solution was filtered through a silica/ celite pad twice. Finally, the filtered solution was concentrated under a reduced pressure, when 95% of solvent got evaporated, *n*-hexane was added slowly into the flask. Further, it was kept stirring at room temperature for 4 hours and the bright greenish solid was collected and dissolved completely in hot toluene, and *n*-hexane was added drop wisely until solid formation and kept stirring at room temperature for an hour, then precipitate was filtered and washed with excess amount of *n*-hexane to obtain the target emitter of 3,11-bis(5-phenylindolo[3,2-a]carbazol-12(5H)-yl)−5,9-dioxa-13b-boranaphtho[3,2,1-de] anthracene DBA-2MICz. Yield: 74 %; bright greenish solid; $^1$H NMR (400 MHz, DMSO-$d_6$) δ: 8.70 (d, $J$ = 8.0 Hz, 1H), 8.62 (d, $J$ = 8.0 Hz, 1H), 8.30-8.35 (dd, $J$ = 8.4, 2.4 Hz, 2H), 8.26 (d, $J$ = 7.2 Hz, 2H), 7.71−7.75 (m, 4H), 7.29−7.67 (m, 19H), 7.16-7.28 (m, 4H), 7.01−7.07 (q, $J$ = 7.2, 8.0 Hz, 2H), 6.67 (d, $J$ = 7.2 Hz, 2H), 6.20 (d, $J$ = 8.0 Hz, 1H), 6.15 (d, $J$ = 8.4 Hz, 1H); HRMS (ESI) $m/z$: Anal. calcd. For $C_{66}H_{39}BN_4O_2$, 930.3166; found, 930.3170.

**Synthesis of DBA-DTMCz**

3,11-dibromo-5,9-dioxa-13b-boranaphtho[3,2,1-de]anthracene DBA-p2Br (1.0 g, 2.33 mmol), 1,3,6,8-tetramethyl-9H-carbazole TMCz (1.09 g, 4.88 mmol), tris(dibenzylideneacetone)dipalladium(0) (0.085 g, 0.09 mmol), 2-dicyclohexylphosphino-2′,4′,6′-triisopropylbiphenyl (0.089 g, 0.18 mmol), sodium *tert*-butoxide (0.62 g, 6.54 mmol) and anhydrous toluene (18 mL) were added in to a dried two neck round bottom flask equipped with a condenser at room temperature. Then, the above mixture kept stirred at 100°C for 6 hours under an inert condition. After complete disappearance of

stating materials, crude mixture was worked up using chloroform (30 mL*2) and water (75 mL). Collected organic layer was dried over anhydrous magnesium sulfate and filtered using a filter paper. Then, collected solution was filtered through a silica/ celite pad twice. Finally, the filtered solution was concentrated under reduced pressure, when 95% of solvent got evaporated, *n*-hexane was added slowly into the flask. Further, it was kept stirring at room temperature for 2.5 hours and the greenish-yellow solid was collected and washed with excess amount of *n*-hexane to obtain the target emitter of 3,11-bis(1,3,6,8-tetramethyl-9H-carbazol-9-yl)−5,9-dioxa-13b-bor-anaphtho[3,2,1-de]anthracene DBA-DTMCz. Yield: 81%; greenish-yellow solid; $^1$H NMR (400 MHz, DMSO-$d_6$) δ: 8.43 (d, $J$ = 7.2 Hz, 2H), 7.80 (s, 4H), 7.47 (m, 3H), 7.33 (d, $J$ = 7.6 Hz, 2H), 7.01 (d, $J$ = 8.0 Hz, 2H), 6.91 (s, 4H), 2.42 (s, 12H, methyl), 1.90 (s, 12H, methyl); HRMS (ESI) $m/z$: Anal. calcd. For $C_{50}H_{41}BN_2O_2$, 712.3261; found, 712.3274.

## Data availability

The authors declare that the data supporting the findings of this study are available within the paper and its supplementary information file.

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

## Acknowledgements
This work was supported by the National Research Foundation of Korea (Grant no. NRF 2019M3D1A2104019).

## Author contributions
H.L. carried out the device study and manuscript writing. R.B. carried out the synthesis and characterization of all materials and supported manuscript writing. H.L. and R.B. designed the molecules and contributed equally to this work. S.M. supported the chemical characterization. C.Y.J. and H.S.L. supported the device-related study. H.L. participated in a major part of the revision process. R.B., C.Y.J., and H.S.L. supported the revision. All the experiments and characterizations were conducted under the supervision of J.H.K.

## Competing interests
The authors declare no competing interests.
