## [Peer Review File · Nature Communications]

Efficient pure blue hyperfluorescence devices utilizing quadrupolar donor-acceptor-donor type of thermally activated delayed fluorescence sensitizersReviewers' Comments:

Reviewer #1:

Remarks to the Author:

In this article, Kwon and coworkers proposed high-efficiency hyperfluorescent (HF) devices can be achieved by hindering the DET process through i) the high kRISC of TADF sensitizer and ii) the shielding of LUMO in D-A-D TADF skeleton. There is no doubt that extraordinary HF devices with maximum external quantum efficiencies (EQEs) of up to 43.9% are demonstrated in this articles; however, I do not think the impact of the finding present in this article is worth to be published in Nature Communications, especially the experimental data is not consistent to the authors' conclusion.

Authors emphasized the importance of blocking the DET process from TADF sensitizer to the final emitter, which can be done by increasing the kRISC and shielding the LUMO of the TADF sensitizer. It is well known that increasing the bulkiness or steric hindrance of TADF sensitizer or final emitter can inhibit the DET process by increasing the intermolecular distance. I would expect similar steric effect is happened in DBA-DTMCz and DBA-DmICz. Thus, it is not convincing that authors claimed the reduction of DET process is due to the shielding of LUMO of TADF sensitizer without giving any experimental proof (only quoting ref. 25). Furthermore, vDABNA is used as the final emitter that is a TADF multi-resonant (MR) emitter. In such case, even DET process is present in HF device, the triplet excitons on the final emitter can still be converted back to single excitons, hence emitting light. As a result, I do not believe the high EQEs in HF devices are due to the high kRISC of TADF sensitizer or a high kFRET from kRISC. A TADF emitter with a high kRISC of $1.2 \times 10^7 \text{ s}^{-1}$, which is comparable to DBA-DTMCz and DBA-DmICz, has been reported by Kaji and coworkers (Nat. Photonics 2020, 14, 643–649). HF device using a fluorescent emitter, TPBe, only resulted in a maximum EQE of 18.7%. From another work by Cui, Adachi, Friend and coworkers (Nat. Photonics 2020, 14, 636–642), a blue HF device with a high EQE of 29.3% is reported. The device is based on a TADF sensitizer having a similar kRISC of $1.5 \times 10^7 \text{ s}^{-1}$ together with TPBe. As a result, I believed that the exceptional high EQEs in HF devices in this work are due to the use of vDABNA, which is a TADF MR emitter. Indeed, the EQEs in HF devices are always higher than that of TADF-only devices. The enhancement of EQEs in HF devices is due to the higher horizontal orientation factor of vDABNA, when compared to those of DBA-DTMCz and DBA-DmICz.

Most importantly, vDABNA is a TADF emitter with a similar delayed lifetime to those of DBA-DTMCz and DBA-DmICz. In Figure 6 and Table S5, the photophysical data in HF films has been provided by authors. But, indeed, there should be two prompt lifetimes and two delayed lifetimes, which are originated from TADF sensitizer and vDABNA. The ignorance of the TADF behavior of vDABNA on calculating the kFRET and kDET is not suitable. In conclusion, the experiment data presented by authors is not sufficient to give such conclusion in this article.

Below are some minor comments:

- 1) Please provide the reduction scan of two TADF sensitizers in CV measurement and calculate the corresponding LUMO levels.
- 2) According El-Sayed rule, the transition between 1CT and 3CT is forbidden; however, both TADF emitters exhibited high kRISCs. Authors explained that the 3LE assists to the RISC process with calculations in Figures S1 and 2. It will be better to indicate the energy differences between all excited states in eV.
- 3) Authors provided energy transfer analysis to calculate the kFRET and kDET. I think it is better to provide a schematic diagram showing all radiative and non-radiative processes happened in HF films. Then, authors should give a better elaboration on how they derive the equations 2 and 3 in the manuscript.

4) As aforementioned, please provide the correct analysis on the determination of τ_p and τ_d of TADF sensitizer in HF films by considering the TADF behavior of ν DABNA.

Reviewer #2:

Remarks to the Author:

This paper reported new quadrupolar donor-acceptor-donor (D-A-D) type TADF sensitizer materials. Two TADF materials revealed high kRISC because of quasi-degenerated HOMO energy levels and additional SOCME value. Therefore, fast kFRET was obtained due to high kRISC. In addition, low kDET value was obtained by donor shielding LUMO structure. As a result, hyperfluorescence device exhibited high EQE of 43.9% with CIE coordinates of (0.12, 0.16). This paper was well analyzed and the EQE result is higher value than reported so far. However, there are major issues should be addressed further before publication.

In Fig. 3(a),(b) and Fig. S3, information of experimental condition should be added such as 10⁻⁵ M toluene solution or 30% doped film.

Additional peak is shown at about 520 nm in the PL spectrum of the 30% doped film in Fig. 3(d). Please explain further detail where this shoulder peak was originated.

The quadrupolar D-A-D type DBA-DTMCz and DBA-DmICz have higher kRISC value than D-A types DBA-TMCz (TMCz-BO) and DBA-mICz. However, it is concerned whether it is appropriate for comparison because the location of substituted donor moiety is different. Isn't it more appropriate to compare a molecule with one substituted donor moiety at the same location at the DBA acceptor?

In order to describe that DBA-DmICz has a higher intensity of ICT absorption than DBA-mICz, it would be better for UV-VIS absorption in Fig. S7(d) to be expressed as an epsilon value.

There is no current efficiency-luminance plot in Fig. 5, but at the sentence in page 10, it is explained there is current efficiency-luminance plot in Fig. 5.

Line 275 in page 10, "Although the solution... 6 nm of deep blue color, ..." should be revised to 7 nm, not 6 nm.

In Fig. S10, horizontal orientation factor of DBA-DTMCz was measured 86%. And, it is noted that lower horizontal orientation factor of DBA-DmICz can be anticipated in page 11. However, it seems to be insufficient to contend that parallel degrees of the direction of TDM in TADF molecules and the molecular longest axis is the main attribution of the high and low horizontal orientation factor. It would be better to address more detail explanations related with horizontal orientation factor of TADF molecule.

In page 11, line 293, the sentence "In addition, ...DBA-DTMCz are studied..." should be revised to "In addition, ...DBA-DTMCz are studied..."

In order to support the high efficiency of EL devices of DBA-DmICz emitter, it is suggested to measure the emitting dipole orientation of DBA-DmICz.

In Fig. S13, please add a bar representing the intensity value of the z-axis EQE percentage.

At schematic diagram for exciton transfer in HF system In Fig. 6(a), the T1 energy of TADF is described lower than the S1 energy of the Final emitter. But, it is expected that the T1 energy of TADF measured in toluene solution is higher than the S1 of v-DABNA on the basis of onset. It needs to be revised. Furthermore, additional measurement data for phosphorescence spectra of 30% doped film is needed to describe the mechanism of Fig. 6(a).

To apply the energy transfer equation of (2), (3) (reference 55, N. Aizawa et al. 2017) which analyze kFRET and kDET by analytical method, it is necessary to measure transient-PL at the areas where only pure TADF emission occurs. In Fig. 6(c) and (d), please indicate the detection wavelength measured for transient-PL. In addition, it would be better to compare the PL spectra of TADF and Final emitter to clarify the areas where only pure TADF emission occurs.

It is suggested to add data measuring the decay of the transient-EL of the devices with a high triplet exciton density to clarify that the Dexter energy transporter was suppressed.

Reviewer #3:

Remarks to the Author:

It has been already published 37% of EQE using out coupling efficiency in dipole TADF materials by in the Wong & Wu group in 2016. (<https://doi.org/10.1002/adma.201601675>). It is achieved high efficiency not only in CT type TADF materials but also in the phosphorescent OLEDs as you mentioned. In addition, as OLED device technology developed, many studies have been published to increase efficiency in Hyperfluorescence (HF) devices.

In this manuscript, this is a study that enhance the transition dipole orientation of materials by Quadrupolar DAD TADF materials and then improve the PLQY and photophysical properties by skeleton structure of increasing LUMO shielding effectness, which is also investigated by the Yasuda Group (ref 29,30).

If you see the direction of the transition dipole orientation (TDO) in Supplementary Figure 11, It might be assumed that DBA-mICz (dipolar material) is expected to have similar TDO directions compare with DBA-DmICz (quadrupolar material). If you are intended to maximize the efficiency by quadrupolar materials to increase device performances due to high PLQY and photophysical properties of materials, you should show the results of DBA-mICz and DBA-DmICz whether PLQY and optical results have been improved due to orientation or structural advantages despite similar orientation.

It is dramatically high EQE in the HF devices and I can agree absolute efficiency of DBA-DTmCz is higer than DBA-DmICz because of the short decay time, fast KFRET and KDET doped with 1% of v-DABNA. However, comparing the efficiency ratio between conventional TADF devices and HF devices, the efficiency ratio of DBA-DTmCz shows a rather low increase rate than DBA-DmICz. Please explain how can interpret the reason.

Response to reviewer's comments

Research Article No: NCOMMS-22-36478

Title: Utilizing Quadrupolar Donor-Acceptor-Donor Type of Thermally Activated Delayed Fluorescence Sensitizer for Achieving Highly efficient and Pure Blue Hyperfluorescence Devices

We are grateful to the reviewers for their useful comments and suggestions. We have revised our present manuscript as per the reviewer's suggestions/comments. The detailed answers to reviewer's comments are as follows. The original comments from the referee are in black. The response to the comment is in blue. The corresponding changes made in the revised manuscript are in red.

Comments & Responses (Reviewer #1):

In this article, Kwon and coworkers proposed high-efficiency hyperfluorescent (HF) devices can be achieved by hindering the DET process through i) the high kRISC of TADF sensitizer and ii) the shielding of LUMO in D-A-D TADF skeleton. There is no doubt that extraordinary HF devices with maximum external quantum efficiencies (EQEs) of up to 43.9% are demonstrated in this articles; however, I do not think the impact of the finding present in this article is worth to be published in Nature Communications, especially the experimental data is not consistent to the authors' conclusion.

Authors emphasized the importance of blocking the DET process from TADF sensitizer to the final emitter, which can be done by increasing the kRISC and shielding the LUMO of the TADF sensitizer. It is well known that increasing the bulkiness or steric hindrance of TADF sensitizer or final emitter can inhibit the DET process by increasing the intermolecular distance. I would expect similar steric effect is happened in DBA-DTMCz and DBA-DmICz. Thus, it is not convincing that authors claimed the reduction of DET process is due to the shielding of LUMO of TADF sensitizer without giving any experimental proof (only quoting ref. 25). Furthermore, vDABNA is used as the final emitter that is a TADF multi-resonant (MR) emitter. In such case, even DET process is present in HF device, the triplet excitons on the final emitter can still be converted back to single excitons, hence emitting light. As a result, I do not believe the high EQEs in HF devices are due to the high kRISC of TADF sensitizer or a high kFRET from kRISC. A TADF emitter with a high kRISC of $1.2 \times 10^7 \text{ s}^{-1}$, which is comparable to DBA-DTMCz and DBA-DmICz, has been reported by Kaji and coworkers (Nat. Photonics 2020, 14, 643–649). HF device using a fluorescent emitter, TPBe, only resulted in a maximum EQE of 18.7%. From another work by Cui, Adachi, Friend and coworkers (Nat. Photonics 2020, 14, 636–642), a blue HF device with a high EQE of 29.3% is reported. The device is based on a TADF sensitizer having a similar kRISC of $1.5 \times 10^7 \text{ s}^{-1}$ together with TPBe. As a result, I believed that the exceptional high EQEs in HF devices in this work are due to the use of vDABNA, which is a TADF MR emitter. Indeed, the EQEs in HF devices are always higher than that of TADF-only devices. The enhancement of EQEs in HF devices is due to the higher horizontal orientation factor of vDABNA, when compared to those of DBA-DTMCz and DBA-DmICz.

Most importantly, vDABNA is a TADF emitter with a similar delayed lifetime to those of DBA-

DTMCz and DBA-DmICz. In Figure 6 and Table S5, the photophysical data in HF films has been provided by authors. But, indeed, there should be two prompt lifetimes and two delayed lifetimes, which are originated from TADF sensitizer and vDABNA. The ignorance of the TADF behavior of vDABNA on calculating the kFRET and kDET is not suitable. In conclusion, the experiment data presented by authors is not sufficient to give such conclusion in this article. Below are some minor comments:

Thank you for your comments and valuable suggestions towards our work, and which helped us to improve our manuscript quality.

1. It is well known that increasing the bulkiness or steric hindrance of TADF sensitizer or final emitter can inhibit the DET process by increasing the intermolecular distance. I would expect similar steric effect is happened in DBA-DTMCz and DBA-DmICz. Thus, it is not convincing that authors claimed the reduction of DET process is due to the shielding of LUMO of TADF sensitizer without giving any experimental proof (only quoting ref. 25).

As reviewers mentioned, increasing bulkiness or steric hindrance of host or emitter can suppress the DET process through increasing the intermolecular distance, and reviewers expected the same effect in this system. In order to confirm this, we have calculated the triplet density distribution of both materials.

From the above figure, we can clearly understand that the triplet density is distributed over the entire molecules. Therefore, DET process cannot be effectively prohibited through their bulkiness or steric hindrance. So, it can be believed that LUMO shielding effect plays a major role in inhibition of DET process in both materials. To strengthen our opinion, rate constant of concentration quenching (k_{CQ}) calculation is performed. Previous report of Yasuda *et al.* demonstrated that the insulating substituents effectively inhibits of the intermolecular electron-exchange interactions, and CQ. (Adv. Mater. 2017, 29, 1604856) Here, the intermolecular electron-exchange interaction can correspond to the Dexter energy transfer in terms of direct electron exchange, and we can deduce that the low CQ indicates low possibility of DET. Here, we have calculated the k_{CQ} values of DBA-mICz, DBA-DmICz, and DBA-DTMCz using 10, 30, 60, and 100% doped in DBFPO film. For the comparison, reported material of MCz-XT was chosen (Adv. Mater. 2017, 29, 1604856) because it consists of the same donor moiety (TMCz) with DBA-DTMCz. D-A type of mICz donor-based DBA-mICz revealed highest k_{CQ} value among three materials. Despite of similar rate constant and same donor moiety (TMCz), D-A-D type of DBA-DTMCz exhibited low k_{CQ} value compared to that of D-A type of MCz-XT. In 60 wt% of MCz-XT, the k_{CQ} value is $3.74 \times 10^5 \text{ s}^{-1}$, while that of DBA-DTMCz is $1.68 \times 10^5 \text{ s}^{-1}$. These results experimentally convinced that D-A-D type of TADF materials are effectively inhibiting the intermolecular electron-exchange interaction (that is DET).

	MCz-XT	DBA-DMTCz	DBA-mICz	DBA-DmICz
Structure				k_{CQ} (s^{-1}) (concentration)	1.08x10 ⁵ (20 wt%)	0.87x10 ⁵ (10%)	1.19x10 ⁵ (10%)	0.67x10 ⁵ (10%)
	2.54x10 ⁵ (40 wt%)	1.12x10 ⁵ (30%)	2.09x10 ⁵ (30%)	0.85x10 ⁵ (30%)
	3.74x10 ⁵ (60 wt%)	1.68x10 ⁵ (60%)	5.42x10 ⁵ (60%)	1.71x10 ⁵ (60%)
	6.18x10 ⁵ (100 wt%)	4.25x10 ⁵ (100%)	8.47x10 ⁵ (100%)	4.15x10 ⁵ (100%)
	Adv. Mater. 2017, 29, 1604856	This work		

Revised Manuscript and Figure:

“On the other hand, the k_{DET} value of *p*MDBA-DI is higher than our materials, which was $2.23 \times 10^5 \text{ s}^{-1}$, despite of the high k_{RISC} of $1.14 \times 10^6 \text{ s}^{-1}$. Such low DET process may be derived of the long intermolecular distance due to inert bulky moiety in the terminal site or LUMO shield D-A-D skeleton of DBA-DmICz and DBA-DTMCz. However, the simulation results indicate that the triplet density of both materials are distributed throughout the skeleton as shown in Supplementary Fig. 16”

Supplementary Fig. 1 The triplet density distribution of (a) DBA-DmICz, and (b) DBA-DTMCz.

Fig. 1 TRPL decay curves and PLQY variation (a) DBA-DmICz, (b) DBA-DTMCz, and (c) DBA-mICz depending on the doping concentration. (d) The plot of PLQY depending on the doping concentration.

...

“Since the LUMO distribution of both two materials were shielded by HOMO from donors, only limited triplet excitons were able to transport via DET channel, resulting low k_{DET} value as illustrated in Fig 6. (b). In order to demonstrate the limited intermolecular electron charge transfer, rate constant of concentration quenching (k_{CQ}) of DBA-mICz, DBA-DmICz and DBA-DTMCz are measured at 10, 30, 60, and 100% doped in DBFPO host film as following the reported equation (4):

$$k_{CQ} = \frac{1}{2} \left(k_{PF} + k_{DF} - 2k_{RISC} - \sqrt{(k_{PF} - k_{DF})^2 - 4k_{ISC}k_{RISC}} \right) \quad (4)$$

Here, the k_{CQ} stands for the effective intermolecular electron-exchange interaction, indicating the long triplet excitons transfer distance and active DET process. Among three TADF materials, DBA-mICz exhibited the highest k_{CQ} value, which is increased from 1.19 to 8.47 $\times 10^5$ s^{-1} from 10 to 100% doped condition. On the other hand, the DBA-DmICz showed low k_{CQ} value of 0.67 to 4.15 $\times 10^5$ s^{-1} . Especially, DBA-DTMCz showed similar k_{CQ} value $\sim 4.25 \times 10^5$ s^{-1} , which is lower than that of D-A type of MCz-XT (6.18 $\times 10^5$ s^{-1}). Presence of same donor, D-A-D type of TADF materials exhibit low k_{CQ} value, which stands for the limited DET process. And this can be due to the lack of LUMO-LUMO overlapping for electron charge transfer. All measured decay curves, PLQY are described in Fig. 7, and calculated rate constants are listed

in Supplementary Table 6.”

...

Supplementary Table S 1 Photophysical properties and rate constants of TADF materials in DBFPO host film at various doping concentration of 10-100%.

	Doping conc. (%)	τ_p (ns)	τ_d (μ s)	Φ_{PL}	Φ_p	Φ_d	k_{PF} (10^7 /s)	k_{DF} (10^5 /s)	k_{ISC} (10^6 /s)	k_{RISC} (10^6 /s)	k_{CQ} (10^5 /s)
DBA-mICz	10	45.9	2.46	0.90	0.64	0.26	2.18	4.07	7.82	-	1.19
	30	45.9	2.04	0.85	0.64	0.21	2.18	4.90	7.60	0.45	2.09
	60	45.9	1.28	0.75	0.63	0.12	2.18	7.81	7.99	-	5.42
	100	43.9	0.91	0.70	0.61	0.09	2.28	10.9	8.72	-	8.47
DBA-DmICz	10	37.3	2.12	0.97	0.71	0.26	2.68	4.72	8.04	-	0.67
	30	37.3	1.94	0.95	0.70	0.25	2.68	5.16	0.81	0.62	0.85
	60	37.3	1.59	0.92	0.69	0.23	2.68	6.29	8.17	-	1.71
	100	37.0	1.37	0.83	0.69	0.14	2.70	7.30	8.41	-	4.15
DBA-DTMCz	10	36.0	0.97	1.00	0.51	0.49	2.78	10.24	13.72	-	0.87
	30	36.0	0.92	0.99	0.51	0.48	2.78	10.91	13.70	2.10	1.12
	60	36.0	0.84	0.97	0.51	0.46	2.78	11.90	13.60	-	1.68
	100	35.2	0.80	0.86	0.50	0.36	2.84	12.52	14.18	-	4.25

2. A TADF emitter with a high k_{RISC} of $1.2 \times 10^7 \text{ s}^{-1}$, which is comparable to DBA-DTMCz and DBA-DmICz, has been reported by Kaji and coworkers (Nat. Photonics 2020, 14, 643–649). HF device using a fluorescent emitter, TPBe, only resulted in a maximum EQE of 18.7%. From another work by Cui, Adachi, Friend and coworkers (Nat. Photonics 2020, 14, 636–642), a blue HF device with a high EQE of 29.3% is reported. The device is based on a TADF sensitizer having a similar k_{RISC} of $1.5 \times 10^7 \text{ s}^{-1}$ together with TPBe. As a result, I believed that the exceptional high EQEs in HF devices in this work are due to the use of vDABNA, which is a TADF MR emitter. Indeed, the EQEs in HF devices are always higher than that of TADF-only devices. The enhancement of EQEs in HF devices is due to the higher horizontal orientation factor of vDABNA, when compared to those of DBA-DTMCz and DBA-DmICz.

We also mentioned that “such high HF device efficiency is supported by good Φ_{FRET} , high PLQY, high θ , and the additional TADF performance of v-DABNA.” in our manuscript. Further, previous report indicates that high device efficiency enhancement in v-DABNA based HF device was mainly due to the improvement of high θ , good Φ_{FRET} , PLQY, and small but additional TADF performance of v-DABNA (Adv. Funct. Mater. 2021, 2110356). We have cited the reference paper in the revised manuscript.

Revised Manuscript:

Especially, DBA-DTMCz utilized HF device exhibited the maximum HF device efficiency among reported v-DABNA based HF devices up to now. Such high device efficiency improvement is supported by high PLQY, improved Φ , and the additional TADF performance of v-DABNA with good Φ_{FRET} as previously reported⁵³.

Although the efficiency enhancement in v-DABNA HF device is mainly due to the improved Φ from v-DABNA, exciton utilization efficiency is also important. As reviewer mentioned in comment, v-DABNA has TADF characteristics, however, it shows inefficient TADF (RISC) performance, which may not fully satisfy the exciton utilizing as much as FRET assisted singlet exciton yield process does. For instance, in TED-HF system, TADF sensitizer acts as triplet exciton distributor and v-DABNA employs more triplet excitons. However, it can be noted that TED-HF system showed limited EQE of 33.5%. (Nature Photonics, 2021, 15, 208-215) In addition, the efficiency enhancement ratio of DBA-BFICz utilized HF device also limited due to its fast ISC process and more populated triplet excitons from heavy atom effect (Adv. Funct. Mater. 2021, 2105805). Our purpose of this study is that enhancing the singlet exciton utilization and improving the device performances while limiting the DET process (triplet exciton employing) through D-A-D type of TADF sensitizers possessing high k_{RISC} and suppressed LUMO-LUMO overlapping.

3. Ignorance of the TADF behavior of vDABNA on calculating the kFRET and kDET is not suitable.

As we mentioned above, the TADF behavior or RISC efficiency of v-DABNA is quite insignificant compared to that of intra-charge transfer (ICT) type of TADF sensitizers. In addition, for calculation of rate constants, we selected 365 nm of excitation wavelength to ensure that the dominant exciton behaviors are coming from the TADF sensitizer. During the excitation or energy transfer procedure, v-DABNA can get triplet excitons via RISC or DET but, doping concentration is small as 1% and it hardly utilizes the triplet excitons due to the inefficient RISC/ISC. Thus, we think that inefficient TADF performance of v-DABNA hardly affects to calculate the k_{DET} . Moreover, previously reported paper also utilized the MR type of TADF material as final dopant and assigned the τ_{p} and τ_{d} in HF film with the same method. (Angew. Chem. Int. Ed. 10.1002/anie.202113206) In the revised manuscript, we mentioned the possibility of influence of TADF behavior from v-DABNA.

Revised Manuscript:

Where k_{PF} and k_{rS} are prompt emission and singlet radiative rate constant, respectively. The k_{DF} and $k_{\text{nr,T}}$ represents the delayed emission and non-radiative triplet decay rate constant, respectively. Although v-DABNA exhibits the TADF performance, constitutes only 1% in the entire HF system and its TADF portion is quite small. Therefore, most of the exciton behaviors reflect the TADF materials, so it can be assumed that TADF performance of v-DABNA hardly affects to calculation of k_{DET} .

#1. Please provide the reduction scan of two TADF sensitizers in CV measurement and calculate the corresponding LUMO levels.

Answer:

Thank you for your comment. As the reviewer requested, we measured the reduction curves of two TADF materials via CV measurement as below figure. As a result, the calculated LUMO level of DBA-DTMCz and DBA-DmICz are 3.24 and 3.21 eV, respectively, which corresponds the same trend with the values described in the original manuscript. In addition, we revised the CV curve in Figure S4 including the reduction curve.

Revised Figure:

Supplementary Fig. 4 (a) Reduction and (b) oxidation cyclic voltammetry (CV) curves of DBA-DmICz and DBA-DTMCz.

#2. According El-Sayed rule, the transition between 1CT and 3CT is forbidden; however, both TADF emitters exhibited high k_{RISC} s. Authors explained that the 3LE assists to the RISC process with calculations in Figures S1 and 2. It will be better to indicate the energy differences between all excited states in eV.

Answer:

Thank you for your comment. As reviewer recommended, we measured the ^3CT of 30% of DBA-DmICz and DBA-DTMCz in DBFPO host film. The calculated values are closely lying for efficient RISC process. In 30% doped of DBA-DmICz, the ΔE_{ST} value is small as 0.05 eV and the energy difference between the ^3LE and ^3CT were also small as 0.07 eV. In 30% doped of DBA-DTMCz, the ΔE_{ST} value is 0.01 eV, and $\Delta E_{3\text{LE}-3\text{CT}}$ is 0.06 eV. Such small energy differences can induce the efficient vibronic coupling between the ^3LE and ^3CT , and transition to ^1CT . We revised the manuscript and figure as follow:

Revised Manuscript and Figure:

Such high k_{RISC} values are attributed to small ΔE_{ST} and enhanced SOC from degenerated dual CT transitions as expected in the NTO calculation. To verify such high k_{RISC} values, we confirmed the characteristics of triplet state in 30% doped film state. Both materials exhibited

a clear ^3CT characteristics of phosphorescence spectra as shown in Supplementary Fig 7. The ΔE_{ST} values in film states of DBA-DmICz and DBA-DTMCz are 0.05 and 0.01 eV, respectively. For efficient RISC process, efficient vibronic coupling (VC) between the ^3LE and ^3CT should be involved. With the closely lying excited states, $\Delta E_{3\text{LE}-3\text{CT}}$ values are 0.07 and 0.06 eV, respectively, which is small enough to efficiently induce the VC. Additionally, large SOCME values between the dual ^1CT and ^3LE accelerates the efficient RISC procedure.

Fig. 4 The energy levels and comparison of RISC and ISC rate constants between bipolar D-A and quadrupolar D-A-D types. (a) DBA-mICz and DBA-DmICz, (b) DBA-TMCz and DBA-DTMCz. The energy levels and rate constants were measured in 30% doped films. The rate constants of DBA-TMCz were reported by Adachi *et al.* using the name of TMCz-BO.⁴²

...

Supplementary Fig. 7 The PL and phosphorescence spectra (77K) in 30% doped film state of (a) DBA-DmICz and (b) DBA-DTMCz.

#3. Authors provided energy transfer analysis to calculate the kFRET and kDET. I think it is better to provide a schematic diagram showing all radiative and non-radiative processes happened in HF films. Then, authors should give a better elaboration on how they derive the

equations 2 and 3 in the manuscript.

Answer:

Thank you for your comment. As the reviewer recommended, we have inserted the energy transfer schematic illustration and corresponding rate constant values happened in HF system in Figure 6. In addition, the derivation of equation 2 and 3 are described in the supporting information section.

Revised Manuscript and Figure:

Fig. 2 Schematic diagram for exciton transfer in HF system. Illustrations with respect to (a) rate constant of RISC process and (b) LUMO and LUMO overlapping. The TRPL measurement TADF, final emitter, and HF system utilizing (c) DBA-DmICz and (d) DBA-DTMCz. The photoexcitation wavelength was 365 nm and the emission wavelength was detected at around 480 nm.

...

The rate constant of FRET and DET can be calculated by³:

The decay rates of singlet (S_1) and triplet (T_1) excitons densities can be described by equation S1 and S2.

$$\frac{dS_1}{dt} = -S_1(k_{r,S} + k_{nr,S} + k_{ISC} + k_{FRET}) + T_1k_{RISC} \quad (S1)$$

$$\frac{dT_1}{dt} = -T_1(k_{r,T} + k_{nr,T} + k_{RISC} + k_{DET}) + S_1k_{ISC} \quad (S2)$$

Equation S1 and S2 can be solved by following biexponential decay

$$S_1, T_1 = A_1 \exp(-k_{PF}t) + A_2 \exp(-k_{DF}t) \quad (S3)$$

Where A_1 and A_2 are the prompt intensity and delayed intensity, respectively.

The decay rates for prompt and delayed fluorescence can be expressed as:

$$(k_{r,s} + k_{nr,s} + k_{ISC} + k_{FRET} - k_{PF})(k_{r,T} + k_{nr,T} + k_{RISC} + k_{DET} - k_{PF}) - k_{ISC}k_{RISC} = 0 \quad (S4)$$

$$(k_{r,s} + k_{nr,s} + k_{ISC} + k_{FRET} - k_{DF})(k_{r,T} + k_{nr,T} + k_{RISC} + k_{DET} - k_{DF}) - k_{ISC}k_{RISC} = 0 \quad (S5)$$

Assuming of,

$$i) k_{PF} \gg k_{DF}$$

$$ii) k_{r,s}, k_{ISC}, k_{FRET} \gg k_{nr,s}, k_{nr,T}, k_{r,T}, k_{DET}$$

Equation S4 and S5 can give:

$$k_{FRET} \approx k_{PF} - k_{r,s} - k_{ISC} \quad (S6)$$

$$k_{DET} \approx \frac{k_{PF}k_{DF} - k_{RISC}(k_{r,s} + k_{FRET})}{k_{r,s} + k_{ISC} + k_{FRET}} - k_{nr,T} \approx k_{DF} - k_{nr,T} - k_{RISC} + \frac{k_{RISC}k_{ISC}}{k_{PF}} \quad (S7)$$

#4. As aforementioned, please provide the correct analysis on the determination of τ_p and τ_d of TADF sensitizer in HF films by considering the TADF behavior of ν DABNA.

Answer:

Thank you for your comment. For the determination of prompt and delayed decay curve of HF system, the exciton behavior of TADF sensitizer is necessary. Thus, detecting of pure TADF sensitizer emission wavelength is required. However, both emission spectra are uni-colored, their emission region is entirely overlapped as below.

Also, with the assistance of high FRET efficiency, HF film hardly exhibited the pure TADF emission only with high intensity of ν -DABNA emission. We tried to detect >500 nm of

emission wavelength several times to measure the pure TADF emission, but decay curves were measured with high noise may be due to the too weak emission intensity. Thus, we detected the high intensity of emission wavelength at around the 480 nm. Instead, we carefully selected the 365 nm of excitation wavelength in order to ensure the dominant excitons behaviors to be generated from the TADF sensitizers as like previously reported uni-colored phosphor sensitized fluorescence system (Nat Commun., 2008, 9, 4990). Since DBFPO host has no light absorption in 365 nm, most of the entire excitons behaviors can be originated from the TADF sensitizers. However, as the reviewer concerned, TADF behavior of *v*-DABNA may affect to the τ_d analysis in HF system because *v*-DABNA also can be photo-excited. But, only 1% of concentration was doped, and its delayed portion and TADF performance is relatively weak compared to that of TADF sensitizers. Thus, we assumed that TADF behavior of *v*-DABNA hardly affect the determination of τ_d . In addition, in previously reported paper also utilized the MR type of TADF emitter as final dopant but, assigned the τ_p and τ_d in HF film. (Angew. Chem. Int. Ed. 10.1002/anie.202113206). We revised the manuscript including TRPL measurement condition as follow.

Revised manuscript:

In order to figure out the reason of such high efficiency enhancement from TADF to HF devices, energy transfer studies were performed. Since the emission spectra of two TADF materials and *v*-DABNA located on the similar wavelength region, detecting pure TADF emission is difficult. Thus, we selected the photoexcitation wavelength of 365 nm to ensure the dominant exciton behaviors from TADF materials.

...

Fig. 6 Schematic diagram for exciton transfer in HF system. Illustrations with respect to (a) rate constant of RISC process and (b) LUMO and LUMO overlapping. The TRPL measurement TADF, final emitter, and HF system utilizing (c) DBA-DmICz and (d) DBA-DTMCz. The photoexcitation wavelength was 365 nm and the emission wavelength was detected at around 480 nm.

Comments & Responses (Reviewer #2):

This paper reported new quadrupolar donor-acceptor-donor (D-A-D) type TADF sensitizer materials. Two TADF materials revealed high kRISC because of quasi-degenerated HOMO energy levels and additional SOCME value. Therefore, fast kFRET was obtained due to high kRISC. In addition, low kDET value was obtained by donor shielding LUMO structure. As a result, hyperfluorescence device exhibited high EQE of 43.9% with CIE coordinates of (0.12, 0.16). This paper was well analyzed and the EQE result is higher value than reported so far. However, there are major issues should be addressed further before publication.

#1. In Fig. 3(a),(b) and Fig. S3, information of experimental condition should be added such as 10⁻⁵ M toluene solution or 30% doped film. Additional peak is shown at about 520 nm in the PL spectrum of the 30% doped film in Fig. 3(d). Please explain further detail where this shoulder peak was originated.

Answer:

Thank you for your comment. As following the reviewer's suggestion, the detailed experimental conditions for the measurement of photophysical properties are included in the main manuscript. Also, bathochromic-shifted and broaden spectrum shape with additional shoulder-like peak in film state can be derived of the concentration induced-aggregation and relaxation effect. Due to 30% of high doping concentration, the concentration induced-aggregation effect can result in the broadening and bathochromic shifting of spectrum shape. Simultaneously, the polar state medium can re-orient the dipole moment and stabilize the excited state energy of fluorescence. Especially, ICT characteristics of TADF materials are known to be more sensitive to the relaxation effect in polar medium. Thus, two TADF materials in polar DBFPO host medium can experience the stabilization of excited state energy level. Therefore, the maximum emission peak shifted to the lower energy region and the spectrum shape can become broaden. This also can be confirmed by the solvatochromism effect in Figure S10. We added the explanation of Figure 3 (a), (b) and Figure S3 as follow:

Revised manuscript:

In order to evaluate the TADF performances of DBA-DmICz and DBA-DTMCz, the measurements of absolute PLQY values and transient PL (TRPL) characteristics were proceeded using 30% doped in DBFPO host film as shown in Fig. 3 (c). **In film PL state, both exhibited the broaden and bathochromic shifted emissions which can be derived of high concentration induced-aggregation effect, and stabilization effect in polar DBFPO host medium. Despite of high doping concentration of 30%, the absolute PLQY values of DBA-DmICz and DBA-DTMCz were maintained high as 0.95 and 0.99, respectively.**

...

Fig. 3 The photophysical measurements. The normalized UV-Vis absorption, PL, and phosphorescence spectra of (a) DBA-DmICz and (b) DBA-DTMCz in toluene solution (10⁻⁵ M). The UV-Vis absorption and PL spectra were measured in 300K, and the phosphorescence spectra were measured in 77K with 30 ms of delaying. (c) TRPL measurement in 30% doped

film of DBFPO host matrix under taken photoexcitation at 365 nm. (d) Spectrum overlapping between normalized 30% doped film PL emissions and absorption epsilon values of ν -DABNA.

...

Supplementary Fig. 3 Phosphorescence spectra of TADF, donor, and acceptor moieties of (a) DBA-DmICz and (b) DBA-DTMCz in 10^{-5} M of toluene solution. The RTPL spectra were measured in 300K and the phosphorescence spectra were measured in 77K with 30 ms of delaying.

#2. The quadrupolar D-A-D type DBA-DTMCz and DBA-DmICz have higher kRISC value than D-A types DBA-TMCz (TMCz-BO) and DBA-mICz. However, it is concerned whether it is appropriate for comparison because the location of substituted donor moiety is different. Isn't it more appropriate to compare a molecule with one substituted donor moiety at the same location at the DBA acceptor?

Answer:

Thank you for your comment. We also understand reviewer's concern, while it is difficult to substitute single donor moiety at the same location in an unsymmetrical pattern. Also, the dependence of donor position on photophysical property and theoretical trend would be negligible. In addition, previously reported papers dealing with the D-A-D quadrupolar type of materials, they also compared them with D-A type of materials with the different donor position like us. Thus, we consider that our photophysical and theoretical analysis also to be an appropriate. (Chem Photo Chem 2020, 4, 82-88, Adv. Optical Mater. 2021, 2101282)

#3. In order to describe that DBA-DmICz has a higher intensity of ICT absorption than DBA-mICz, it would be better for UV-VIS absorption in Fig. S7(d) to be expressed as an epsilon value.

Answer:

Thank you for your comment. As reviewer suggested, we revised the y-axis of UV absorption graph to the epsilon value by measuring the UV absorption spectra of DBA-mIC and DBA-DmICz in 10^{-4} M of condition. In addition, we revised the experimental condition and figure S8 (d) in supporting information as follow:

Revised Figure:

Supplementary Fig. 8 (a) The UV-Vis absorption, PL, and phosphorescence spectra of DBA-mICz in toluene state. (b) TRPL measurement in 30% doped film state and (c) CV measurement of DBA-mICz. (d) The comparison of UV-Vis absorption spectra between DBA-mICz and DBA-DmICz at a concentration of 1×10^{-4} M.

#4. There is no current efficiency-luminance plot in Fig. 5, but at the sentence in page 10, it is explained there is current efficiency-luminance plot in Fig. 5.

Answer:

Thank you for your comment. We deleted the expression of current efficiency-luminance plot in manuscript.

Revised manuscript:

All the plots of the current density-voltage-luminance (J - V - L) characteristics, external quantum efficiency versus luminance (EQE- L), and the electroluminescence (EL) spectra at $1,000 \text{ cd/m}^2$ are illustrated in Fig. 5.

#5. Line 275 in page 10, "Although the solution... 6 nm of deep blue color, ..." should be revised to 7 nm, not 6 nm.

Answer:

Thank you for pointing out the mistakes. We revised the manuscript where the reviewer pointed out, and carefully checked the entire sentences in the manuscript.

Revised manuscript:

Although the solution PL of DBA-DmICz in toluene emitted the 7 nm of deep blue color, the EL spectrum in polar medium of DBFPO (host) revealed further bathochromic shifted emission due to stronger stabilization characteristics.

#6. In Fig. S10, horizontal orientation factor of DBA-DTMCz was measured 86%. And, it is noted that lower horizontal orientation factor of DBA-DmICz can be anticipated in page 11. However, it seems to be insufficient to contend that parallel degrees of the direction of TDM in TADF molecules and the molecular longest axis is the main attribution of the high and low horizontal orientation factor. It would be better to address more detail explanations related with horizontal orientation factor of TADF molecule.

Answer:

Thank you for your valuable suggestions. In general, the horizontal orientation factor can be increased when the direction of transition dipole moment (TDM) is aligned horizontally with the plane of the material. The TDM can be expressed by following expression:

$$\vec{p}_{TDM} \propto \left\langle \psi_{\text{HOMO}} \left| \vec{r}_{12} \right| \psi_{\text{LUMO}} \right\rangle$$

The TADF materials are usually comprised of the donor and the acceptor skeleton and have charge transfer state of emission pathway. Since the HOMO is mainly located on the donor and the LUMO is dispersed on the acceptor, the direction of TDM often closely coincides with the long axis of donor and acceptor skeleton. (Front. Chem. 8:750. doi: 10.3389/fchem.2020.00750) Therefore, many TADF materials have been reported to have high horizontal orientation factor by increasing the long axis between donor and acceptor. (Adv. Optical Mater. 2018, 6, 1701340, Adv. Mater. 2020, 32, 2004083) Here, the TDM of both materials were also located along with the longest axis of the D-A-D direction, which is the CT transition direction. Thus, the alignment of TDM in long axis of TADF can enhance the orientation factor. In DBA-DmICz, the acceptor was slightly bent, and the TDM was located with slightly different directions. Thus, we anticipated the lower horizontal orientation, and the measured value was also 0.79. With following the reviewers' advice, we included the detailed explanation and revised the manuscript as follow.

Revised manuscript:

In general, high light out coupling can be obtained when the direction of transition dipole moment (TDM) is aligned horizontally with the plane of the material. And the direction of TDM often closely coincides with the longest axis of donor and acceptor skeleton in TADF material. Thus, the TDMs of DBA-DmICz and DBA-DTMCz were studied to confirm the high Θ value. As expected, the direction of TDM of DBA-DTMCz lied parallel to the longest axis of D-A-D linking as illustrated in Supplementary Fig. 11 (b).

#7. In page 11, line 293, the sentence "In addition, ...DBA-DTMCz are stidied..." should be revised to "In addition, ...DBA-DTMCz are studied...".

Answer:

We have revised the word where the reviewer pointed out, and carefully checked the entire sentences in the manuscript.

Revised manuscript:

In addition, the transition dipole moments of DBA-DmICz and DBA-DTMCz are studied to confirm the high θ value.

#8. In order to support the high efficiency of EL devices of DBA-DmICz emitter, it is suggested to measure the emitting dipole orientation of DBA-DmICz.

Answer:

Thank you for your comment. As the reviewer recommended, we additionally measured the dipole orientation factor of DBA-DmICz, and the calculated value was 0.79. We revised the manuscript and Figure S12 as follow:

Revised manuscript and Figure:

Supplementary Fig. 12 Measurement of horizontal orientation ratio of (a) DBFPO: 30% of DBA-DmICz, (b) DBFPO: 30% of DBA-DTMCz, and (c) DBFPO: 30% of DBA-DTMCz: 1% of v-DABNA.

Since the light mainly emits and propagates perpendicular to the direction of transition dipole moment of emitter, light out coupling efficiency can be affected depending on the θ value. Thus, the θ measurement was proceeded using the previously reported method.

...

Thus, the lower θ value may be obtained. As expected, DBA-DmICz and DBA-DTMCz showed high θ of 0.79 and 0.86, respectively, as shown in Supplementary Fig. 10 (a) and (b).

#9. In Fig. S13, please add a bar representing the intensity value of the z-axis EQE percentage.

Answer:

Thank you for your suggestion. As the reviewer suggested, we included the intensity value of the z-axis EQE value and revised the Figure S15.

Revised Figure:

Supplementary Fig. 15 Calculated theoretical EQE values depending on the PLQY and horizontal orientation factor.

#10. At schematic diagram for exciton transfer in HF system In Fig. 6(a), the T1 energy of TADF is described lower than the S1 energy of the Final emitter. But, it is expected that the T1 energy of TADF measured in toluene solution is higher than the S1 of v-DABNA on the basis of onset. It needs to be revised. Furthermore, additional measurement data for phosphorescence spectra of 30% doped film is needed to describe the mechanism of Fig. 6(a).

Answer:

Thank you for your valuable suggestion. As the reviewer advised, we matched the energy level of individual materials at schematic diagram of figure 6 (a), and revised the figure as below. In addition, we measured the 30% doped film state of phosphorescence energy level as mentioned in answer for question 2 from reviewer #1. Following the reviewers advices, the experimentally measured energy levels of TADF materials enabled us to explain the interconnection between ^1CT , ^3CT and ^3LE , and verify the short delayed decay time and high RISC rate constant of TADF materials. These contents were included in Figure 4, Figure S7 and manuscript. (please check answer for Q2 in reviewer#1)

Revised Figure:

Fig. 6 Schematic diagram for exciton transfer in HF system. Illustrations with respect to (a) rate constant of RISC process and (b) LUMO and LUMO overlapping. The TRPL measurement TADF, final emitter, and HF system utilizing (c) DBA-DmICz and (d) DBA-DTMCz. The photoexcitation wavelength was 365 nm and the emission wavelength was detected at around 480 nm.

#11. To apply the energy transfer equation of (2), (3) (reference 55, N. Aizawa et al. 2017) which analyze kFRET and kDET by analytical method, it is necessary to measure transient-PL at the areas where only pure TADF emission occurs. In Fig. 6(c) and (d), please indicate the detection wavelength measured for transient-PL. In addition, it would be better to compare the PL spectra of TADF and Final emitter to clarify the areas where only pure TADF emission occurs.

Answer:

Thank you for your comment. As the reviewer pointed out, it is required to detect the only pure TADF emission wavelength for calculating the energy transfer. However, as shown in the figure below, two TADF sensitizer materials and v-DABNA acceptor material exhibit the uni-colored emission wavelength, thus, it is hard to distribute clearly. Also, with the assistance of high FRET efficiency, HF system film showed nearly v-DABNA emission with small amount of TADF portion. We also tried to detect >500 nm of emission wavelength to measure the pure TADF emission, but decay curves were incorrectly measured with high noise may due to the too weak emission intensity. Thus, we detected the high intensity of emission wavelength at around the 480 nm. However, in HF system, only 1% of v-DABNA was doped, and we used the 365 nm of excitation wavelength to ensure the predominant exciton harvesting from the

TADF donor materials. Thus, most of the excitons were generated from the TADF sensitizer materials and the decay curves were also predominantly composed of the exciton behaviors from TADF sensitizer. In addition, in previously reported paper, Heimel, P. *et al.* also detected the phosphor and final emitter combined emission wavelength for analyzing exciton dynamics in unicolor phosphor-sensitized fluorescence system (Nat Commun., 2008, 9, 4990). Thus, we think our measurement condition can be acceptable. In addition, we revised the manuscript including TRPL measurement condition.

Revised manuscript:

In order to figure out the reason of such high efficiency enhancement from TADF to HF devices, energy transfer studies were performed. Since the emission spectra of two TADF materials and v-DABNA located on the similar wavelength region, detecting pure TADF emission is difficult. Thus, we selected the photoexcitation wavelength of 365 nm to ensure the dominant exciton behaviors from TADF materials.

...

Fig. 6 Schematic diagram for exciton transfer in HF system. Illustrations with respect to (a) rate constant of RISC process and (b) LUMO and LUMO overlapping. The TRPL measurement TADF, final emitter, and HF system utilizing (c) DBA-DmICz and (d) DBA-DTMCz. The photoexcitation wavelength was 365 nm and the emission wavelength was detected at around 480 nm.

#12. It is suggested to add data measuring the decay of the transient-EL of the devices with a high triplet exciton density to clarify that the Dexter energy transporter was suppressed.

Answer:

Thank you for your suggestion. To verify the superiority of D-A-D type of TADF materials for alleviating DET, the additional experimental evidence is required. And we think that's why reviewer also recommended to measure the transient-EL in high triplet exciton region. As you can see our answer to reviewer #1, we evaluated the additional TADF characteristics of three materials depending on the doping concentration of 10, 30, 60, and 100% to confirm the short triplet exciton diffusion length of D-A-D type TADF materials compared to (similar) D-A type TADF materials. Here, as I answered to Q11, TADF generates almost all excitons at 365 nm of excitation wavelength. And the densely doped TADF materials have closer intermolecular distance between and generate more triplet excitons, which can correspond to the high triplet exciton density condition. Since their concentration quenching rate constant (k_{CQ}) in high

doping concentration was lower than similar structure of D-A TADF, triplet exciton diffusion length of D-A-D type TADF also can be shorter than D-A type.

	MCz-XT	DBA-mICz	DBA-DmICz	DBA-DMTCz
Structure				k_{CQ} (s^{-1}) (neat film)	6.18×10^5	8.47×10^5	4.15×10^5	4.25×10^5
	Adv. Mater. 2017, 29, 1604856	This work		

Thus, the triplet excitons hardly move to the nearby molecules via DET, which we can expect that the DET in high triplet exciton density can also be lower than other D-A type TADF materials. In addition, since the TRPL results are well corresponded to TREL, we think that our TRPL results can satisfy as the answer to your questions.

Comments & Responses (Reviewer #3):

It has been already published 37% of EQE using out coupling efficiency in dipole TADF materials by in the Wong & Wu group in 2016. (<https://doi.org/10.1002/adma.201601675>). It is achieved high efficiency not only in CT type TADF materials but also in the phosphorescent OLEDs as you mentioned. In addition, as OLED device technology developed, many studies have been published to increase efficiency in Hyperfluorescence (HF) devices.

In this manuscript, this is a study that enhance the transition dipole orientation of materials by Quadrupolar DAD TADF materials and then improve the PLQY and photophysical properties by skeleton structure of increasing LUMO shielding effectness, which is also investigated by the Yasuda Group (ref 29,30).

If you see the direction of the transition dipole orientation (TDO) in Supplementary Figure 11, It might be assumed that DBA-mICz (dipolar material) is expected to have similar TDO directions compare with DBA-DmICz (quadrupolar material). If you are intended to maximize the efficiency by quadrupolar materials to increase device performances due to high PLQY and photophysical properties of materials, you should show the results of DBA-mICz and DBA-DmICz whether PLQY and optical results have been improved due to orientation or structural advantages despite similar orientation.

Answer:

Thank you for your comments. As for the reviewer questions, we additionally present the TADF device results of 30 and 40% of DBA-mICz in the same device configuration as below. The maximum EQE of 30% doped DBA-mICz was 18.6% and the EL maximum was located at 467 nm. The device performance is poorer than that of 30% doped DBA-DmICz. As for the material and device efficiency improvements of quadrupolar type of DBA-DmICz, it can be attributed to both enhancement of PLQY (also with TADF performance) and horizontal orientation. As we mentioned in the manuscript, the quasi-degenerated HOMO energy levels and dual CT transition can improve the TADF performance, and the intensified ICT absorption can enhance the prompt and total PLQY. Here, the intensified ICT absorption can be confirmed from the Figure S7 (d), and the prompt and total PLQY was also increased from 64 to 70% and 85 to 95%, respectively. In addition, horizontal orientation factor also can be increased. The horizontal orientation of 30% doped DBA-DmICz was measured to be 0.79. Unfortunately, we could not get the horizontal orientation value of DBA-mICz due to the lack of material amount, but we can guess that the value can be around 0.70 considering the device evaluation results and EQE simulation results in Figure S13. Also, Park *et al.* also reported the enhanced horizontal orientation value of D-A-D type TADF compared to that of D-A type. (Adv. Optical Mater. 2021, 2101282). The horizontal orientation value of OBO-I (D-A type) was 75% and the D-A-D type of OBO-II was measured to be 83%. We have added the device results of DBA-mICz in the revised manuscript as per reviewer's suggestion.

Revised manuscript and Figure:

Subsequently, the intensified ICT absorption can enhance the prompt and total PLQY of DBA-mICz to DBA-DmICz. The prompt PLQY was increased from 64 to 70%, and the total PLQY was enhanced from 85 to 95%. Considering that the mostly reported k_{DET} range of $\sim 10^5$ s⁻¹,~

....

With 99% of absolute PLQY value, DBA-DTMCz was able to achieve such high EQE_{max} value among the reported D-A-D types of blue TADF materials. In order to verify the high performance of D-A-D type of both TADF materials, the device performances of D-A type of DBA-mICz were also evaluated. In the same device configuration, 30% of DBA-mICz exhibited the 18.6% of EQE_{max} value, and the maximum EL peak was 467 nm. Thus, it can be confirmed that TADF device performance of D-A-D skeleton is better than that of D-A skeleton. The device performances of DBA-mICz are illustrated in the Supplementary Fig. 13 and Supplementary Table 5.

Supplementary Fig. 12. Device performances of DBA-mICz with 30 and 40% of doping concentration.

Supplementary Table S 2 TADF Device performances of DBA-mICz with 30 and 40% of doping concentration in DBFPO host device.

	Current		Efficiency (Cd/A) (Max / 1,000 cd/m ²)	EQE (%) (Max / 1,000 cd/m ²)	Max emission peak (nm) ^(c)	FWHM (nm) ^(c)	CIE coordinates (1,000 cd/m ²)
	Turn on voltage ^(a) (V)	Driving voltage ^(b) (V)					
30% DBA-mICz	3.2	5.3	23.9 / 17.6	18.6 / 14.6	467	67	(0.15, 0.18)
40% DBA-mICz	3.2	5.2	23.8 / 19.5	17.3 / 14.9	471	68	(0.15, 0.21)

(a) Turn on voltage at 1 cd/m², (b) Driving voltage at 1,000 cd/m², (c) Measured at 1,000 cd/m².

It is dramatically high EQE in the HF devices and I can agree absolute efficiency of DBA-DTmCz is higher than DBA-DmICz because of the short decay time, fast KFRET and KDET doped with 1% of v-DABNA. However, comparing the efficiency ratio between conventional TADF devices and HF devices, the efficiency ratio of DBA-DTmCz shows a rather low increase rate than DBA-DmICz. Please explain how can interpret the reason.

Answer:

Thank you for your comment. As reviewer pointed out, The DBA-DTMCz based HF system exhibited the high k_{FRET} with low k_{DET} , which can expect the higher HF device efficiency. However, their device improvement rate was similar (1.19 time). The device performance improving rate of DBA-DTMCz is limited than our expectation. Unfortunately, we cannot figure out the exact reason for low improvement rate but, it is clear that their enhancement rates are higher than previously reported value with D-A type TADF emitter having similar k_{FRET} .

In previous reported *p*MDBA-DI material based TADF device (Adv. Funct. Mater. 2021, 2110356), which exhibited the 483 nm of maximum EL wavelength and similar k_{FRET} of $4.14 \times 10^7 \text{ s}^{-1}$ with DBA-DmICz (3.82×10^7) and DBA-DTMCz (4.26×10^7). However, the k_{DET} of *p*MDBA-DI is higher ($2.23 \times 10^5 \text{ s}^{-1}$) than DBA-DmICz and DBA-DTMCz (1.94 and $1.70 \times 10^5 \text{ s}^{-1}$), and the *p*MDBA-DI HF device improvement is limited to 1.14 time. But our current DBA-DmICz and DBA-DTMCz exhibited 1.19 times of improvement. Although they unexpectedly exhibited the similar enhancement ratio value between TADF and HF device, it is clear that the HF device can be improved more in DBA-DmICz and DBA-DTMCz HF system with lower k_{DET} .

We are grateful to the reviewers for their valuable suggestions and comments given, and which made us to improve the manuscript quality a lot. We thank all reviewers for their interest in our current work.

Thank you,

Best Regards,

Professor. Jang Hyuk Kwon

REVIEWER COMMENTS

Reviewer #1 (Remarks to the Author):

In the revised manuscript, I am happy that most of my questions have been solved with additional experiments by the authors; however, I am still not satisfied for the ignorance of TADF properties of vDABNA for the calculation of kDET. Although authors cited a reference that others also assigned the τ_p and τ_d in HF film with the same method, one of the foci in the paper is the suppression of kDET from TADF to final emitter which resulted in a high EQE. Therefore, I would like to know when vDABNA is replaced by a blue fluorescent emitter with no TADF properties, e.g TBPe, what will be the EQE and kDET? I would expect similar high EQE and small kDET can be achieved. If not, the high EQE obtained in this paper is indeed due to the TADF properties of vDABNA or its high orientation, but not because of the suppression of kDET. In that case, the finding/hypothesis in this article may be wrong.

Reviewer #2 (Remarks to the Author):

This work is ready for publication

Reviewer #3 (Remarks to the Author):

Thank you for the detailed explanation in the revised manuscript. I agree to publish the paper with sufficient explanation.

Response to reviewer's comments

Research Article No: NCOMMS-22-36478A

Title: Utilizing Quadrupolar Donor-Acceptor-Donor Type of Thermally Activated Delayed Fluorescence Sensitizer for Achieving Highly efficient and Pure Blue Hyperfluorescence Devices

We have revised our present manuscript as per the reviewer's suggestions/comments. The detailed answer to reviewer's comments are as follows. The corresponding changes made in the revised manuscript are in blue.

Reviewer #1

In the revised manuscript, I am happy that most of my questions have been solved with additional experiments by the authors; however, I am still not satisfied for the ignorance of TADF properties of *v*DABNA for the calculation of k_{DET} . Although authors cited a reference that others also assigned the τ_p and τ_d in HF film with the same method, one of the foci in the paper is the suppression of k_{DET} from TADF to final emitter which resulted in a high EQE. Therefore, I would like to know when *v*DABNA is replaced by a blue fluorescent emitter with no TADF properties, e.g TBPe, what will be the EQE and k_{DET} ? I would expect similar high EQE and small k_{DET} can be achieved. If not, the high EQE obtained in this paper is indeed due to the TADF properties of *v*DABNA or its high orientation, but not because of the suppression of k_{DET} . In that case, the finding/hypothesis in this article may be wrong.

Answer:

Thank you for your comment. As the reviewer recommended, we calculated the k_{DET} from TADF sensitizer to the final emitter of TBPe with no TADF performance in order to demonstrate that the LUMO shielded D-A-D type of TADF sensitizer hindered the electron transfer and DET process. For comparison, we additionally selected the D-A type of TADF material, *p*MDBA-DI (Adv. Funct. Mater. 2021, 2110356) since it exhibits the similar emission wavelength (480 nm, 20% in DBFPO host) and high k_{RISC} value ($1.12 \times 10^6 \text{ s}^{-1}$) as like DBA-DTMCz. And, we built the HF system with 20% of TADF sensitizers and 1% of TBPe in DBFPO host medium. Since the TBPe has low absorption at around 340 nm, we selected the excitation wavelength to 340 nm.

In 20% of TADF system, both DBA-DTMCz and *p*MDBA-DI showed short delayed lifetime of 0.96 and 1.96 μs , respectively. When 1% of TBPe was doped, the delayed portion is decreased and the delayed lifetime also reduced to 0.58 and 0.89 μs , respectively (as shown in the figure above). With using the same equation in the manuscript, we calculated individual k_{DET} values, and they were obtained as 2.48 and $3.12 \times 10^5 \text{ s}^{-1}$ for DBA-DTMCz and *p*MDBA-DI respectively. Compared to the calculated k_{DET} value with 1% of *v*-DABNA ($1.70 \times 10^5 \text{ s}^{-1}$), the k_{DET} value of TBPe is large, and which can be due to the smaller inert groups in terminal site of TBPe. The TBPe is only shielded by two pairs of tert-butyls, while, *v*-DABNA is surrounded by inert and bulky phenyl groups in the terminal sites. Thus, the k_{DET} can be larger. Further, it can be noticed from the above figure, TBPe shows smaller spectral overlap compared to that of *v*-DABNA. Therefore, the FRET channel is less activated and the singlet excitons are much prone to transfer to ISC process when the TBPe was utilized as final emitter. However, it is definitely clear that DBA-DTMCz obtained the lower k_{DET} than that of *p*MDBA-DI only with *v*-DABNA, but also with TBPe final emitter. The detailed values are summarized in the below table.

	Dopant concentration (%)	τ_{p} (ns)	τ_{d} (μs)	k_{FRET} ($10^7/\text{s}$)	k_{DET} ($10^5/\text{s}$)
DBA-DTMCz	0	36.1	0.96		
	1	14.9	0.58	3.38	2.48
pMDBA-DI	0	30.0	1.96		
	1	15.9	0.89	2.97	3.12

With the energy transfer study, we fabricated the TADF and HF devices with TBPe to elucidate the high efficiency in DBA-DTMCz with lower DET process. Initially, 20% of DBA-DTMCz and *p*MDBA-DI TADF devices are constructed, and they exhibited the EQE_{max} values of 35.9 and 32.4% along with the emission maxima of 476 and 479 nm, respectively. Then the HF devices are fabricated with 1% of TBPe in the same device configuration. As a result, DBA-DTMCz sensitized HF device showed the 24.0% of EQE_{max} values, while *p*MDBA-DI sensitized HF device showed only 18.9% as shown in the below figure.

Unlike *v*-DABNA based HF devices, the EQE_{max} in HF device was further decreased. This is due to the insufficient FRET efficiency of TBPe. As reviewer can notice in the figure above, the area of spectral overlap between TADF sensitizers and TBPe is almost half than that of with *v*-DABNA. The FRET was insufficient to transfer the singlet excitons to TBPe, and which resulted in the unclear shape of combined EL spectra and the unexpected lower HF efficiency. Such lower efficiency result can be supported by the 5Cz-Trz (Nature Photonics 2020, 636-642) and HDT-1 (Adv. Optical Mater. 2022, 2200682) based TBPe HF device results. When compare the EL spectrum shape, we can guess that the 5Cz-Trz TADF has higher FRET efficiency, and it achieved the higher HF device efficiency than that of HDT-1 based HF device. (5Cz-Trz TADF/HF: 29.3% / 24% HDT-1 TADF/HF: 25% / 16%). Also, both materials (5Cz-Trz and HDT-1) have LUMO shielded multi-D-A type of TADF and may have advantage of less DET process as like our materials. In addition, as we mentioned in the manuscript and our previously reported paper (Adv. Funct. Mater. 2021, 2110356), the major contribution of high efficiency improvement is derived of higher horizontal orientation factor of *v*-DABNA. While, the TBPe has lower horizontal orientation, and the out-coupling efficiency can be reduced. However, the DBA-DTMCz sensitized HF device exhibited the higher efficiency and lower k_{DET} than that of *p*MDBA-DI, which can support our result. Since TBPe has no any TADF performance, the HF/TADF efficiency ratio gap became much clear in TBPe HF system.

Although we obtained the poor device efficiency with TBPe due to lower FRET efficiency, energy transfer study results and device efficiency results can demonstrate our opinion that D-A-D type of TADF sensitizer can achieve the efficient HF devices by suppressing the DET process. In addition, we revised the manuscript including TRPL measurement result with TBPe and calculated rate constant values.

Revised manuscript and Figure:

“On the other hand, the k_{DET} value of *p*MDBA-DI is higher than our materials, which are $2.23 \times 10^5 \text{ s}^{-1}$, despite of the high k_{RISC} of $1.14 \times 10^6 \text{ s}^{-1}$. **In order to confirm the lower k_{DET} value**

with DBA-DTMCz, we additionally performed the energy transfer study with 2,5,8,11-tetra-tert-butylperylene (TBPe), and compared with D-A type of *p*MDBA-DI. Since the TBPe is pure fluorescent blue material, we can avoid the influence of TADF performance from final emitter. With 20% of TADF sensitizers and 1% of TBPe, the k_{DET} values are obtained to 2.48 and $3.12 \times 10^5 \text{ s}^{-1}$ in DBA-DTMCz and *p*MDBA-DI HF system, respectively. Although the absolute values are slightly increased, DBA-DTMCz exhibited the lower k_{DET} value with TBPe. All the calculated TRPL results are illustrated in Supplementary Fig. 16 and Supplementary Table 6-7.

Such low DET process may be derived of the long intermolecular distance due to the presence of bulky moiety at the terminal site or LUMO shielded D-A-D skeleton of DBA-DmICz and DBA-DTMCz.”

Supplementary Fig. 1 The TRPL measurement of TADF and HF system utilizing (a) DBA-DTMCz and (b) *p*MDBA-DI. The photoexcitation wavelength was 340 nm.

Supplementary Table S 7 TRPL decay lifetime and calculated rate constant of FRET and DET in 20% of DBA-DTMCz and *p*MDBA-DI HF system with 1% of TBPe.

	Dopant concentration (%)	τ_p (ns)	τ_d (μ s)	k_{FRET} (10^7 /s)	k_{DET} (10^5 /s)
DBA-DTMCz	0	36.10	0.96		
	1	14.90	0.58	3.38	2.48
p MDBA-DI	0	30.0	1.96		
	1	15.90	0.89	2.97	3.12

We are grateful to the reviewers for their positive feedback and comments given, and which made us to improve the manuscript quality a lot. We thank all reviewers for their interest in our current wok.

Thank you,

Best Regards,

Professor. Jang Hyuk Kwon

REVIEWERS' COMMENTS

Reviewer #1 (Remarks to the Author):

The manuscript can be accepted in its current form.